# Estimation of joint torque in dynamic activities using wearable A-mode ultrasound

Yichu Jin [1], Jonathan T. Alvarez [1], Elizabeth L. Suitor[1], Krithika Swaminathan [1], Andrew Chin [1], Umut S. Civici [1], Richard W. Nuckols[1,2], Robert D. Howe [1] & Conor J. Walsh [1] ✉

The human body constantly experiences mechanical loading. However, quantifying internal loads within the musculoskeletal system remains challenging, especially during unconstrained dynamic activities. Conventional measures are constrained to laboratory settings, and existing wearable approaches lack muscle specificity or validation during dynamic movement. Here, we present a strategy for estimating corresponding joint torque from muscles with different architectures during various dynamic activities using wearable A-mode ultrasound. We first introduce a method to track changes in muscle thickness using single-element ultrasonic transducers. We then estimate elbow and knee torque with errors less than 7.6% and coefficients of determination ($R^2$) greater than 0.92 during controlled isokinetic contractions. Finally, we demonstrate wearable joint torque estimation during dynamic real-world tasks, including weightlifting, cycling, and both treadmill and outdoor locomotion. The capability to assess joint torque during unconstrained real-world activities can provide new insights into muscle function and movement biomechanics, with potential applications in injury prevention and rehabilitation.

The human body constantly undergoes dynamic mechanical loading during movement and locomotion. However, excessive loading within the body can lead to fatigue and increase the risk of injury[1,2]. Quantifying internal mechanical loads within the musculoskeletal system can thus be valuable for risk assessment and injury prevention in sports and ergonomics[2–4]. Post-injury or surgery, the monitoring of loads experienced by the damaged tissue remains important for facilitating optimal tissue healing[5]. Additionally, in the field of robotics, tracking muscle and joint loads shows potential for developing effective wearable assistive robots for clinical rehabilitation and performance augmentation[6,7]. Despite the wide range of applications, measuring in vivo mechanical loads within the musculoskeletal system remains challenging with existing technologies, especially during unconstrained dynamic movements[3,8].

Research has explored quantifying biomechanical loads on joints and on the various tissues comprising these joints (e.g.,

muscles, tendons, bones). At the tissue level, surgically implanted force transducers have been used to measure forces in muscle-tendon units[9,10] and bones[11]. However, the invasiveness of these implants limits their applicability in human studies. Shear wave tensiometry estimates superficial tendon forces by mechanically exciting the tendon and measuring resultant shear wave speeds[12]. Despite the promising results in estimating Achilles and patellar tendon forces[12,13], its applicability to less prominent or deep tendons has yet to be demonstrated. At the joint level, isokinetic dynamometers can directly measure joint torques and assess the force-generation capacity of an agonist muscle group[14]. However, these lab-based equipment limit joint torque measurements to controlled muscle contractions in a fixed testing setup. Modeling techniques (e.g., inverse dynamics, computational musculoskeletal models)[15,16] have been commonly used to indirectly estimate both muscle force and joint torque. Specifically, inverse dynamics can reliably estimate joint

[1]John A. Paulson School of Engineering and Applied Sciences, Harvard University, Cambridge, MA, USA. [2]Mechanical and Industrial Engineering, University of Massachusetts Lowell, Lowell, MA, USA. ✉e-mail: walsh@seas.harvard.edu

torques within a rigid body model using motion and external force measurements. With additional optimization and biomechanical constraints, musculoskeletal models can further estimate individual muscle forces. Despite these simulation models' wide use in biomechanical studies, they rely on intensive computation with a combination of measurements (e.g., motion-capture systems, force platforms) that are yet largely constrained to research environments[8]. Surface electromyography (EMG) has been used as a wearable alternative for studying muscle force and joint torque[17,18]. However, EMG measures the electrical activation of a muscle, representing neurological input rather than the muscle's mechanical output, and its measurements are hence susceptible to electrical artifacts like sweat or humidity-induced impedance mismatch and neurological conditions like fatigue or neuromotor disorders[19,20].

Recently, researchers have begun to explore various measures of muscle deformation, inspired by its direct coupling with muscle force production[21]. Mechanomyography (MMG) is often considered as the mechanical counterpart to EMG, and it estimates muscle mechanical loads by measuring the lateral oscillations elicited from contracting muscle fibers. However, like EMG, the MMG signal alone is susceptible to crosstalk contamination from nearby muscles and motion artifacts from limb movements[22,23]. Force myography (FMG) captures the muscle geometry change by measuring pressure variations on the skin that occurs from muscle bulging. By using a band of pressure sensors wrapped around the limb, FMG captures the net shape change from all muscles within the limb, often including agonist and antagonist muscle pairs[24]. More recently, ultra-sensitive soft strain sensors have been adhered to skin to capture the localized skin deformation from the underlying muscle bulge, which has been found to positively correlate with changes in joint torque[25]. However, this surface-level measurement cannot decouple deformations from superficial and deep muscles. Muscle specificity is desirable because muscles at different depths can have different functionalities. For instance, the gastrocnemius muscle serves both as an ankle plantar flexor and a knee flexor, whereas the soleus underneath functions only as an ankle plantar flexor[26]. To achieve muscle specificity, brightness mode (B-mode) ultrasound imaging has been used to extract changes in architectural parameters of specific muscles. Parameters such as muscle thickness, fascicle length, and pennation angle have been shown to correlate with muscle force or joint torque measurements during static muscle contractions[27–29] and dynamic functional movements[30–32]. However, despite recent advances in both microfabrication of miniaturized wearable imaging systems[33,34] and automated parameter extraction algorithms[35,36], B-mode ultrasound remains primarily limited to research settings due to its hardware and computational requirements.

Amplitude mode (A-mode) ultrasound offers a potential lightweight solution for muscle-specific deformation measurements. As a simplified alternative to B-mode, A-mode ultrasound can track one-dimensional muscle deformation, like muscle thickness, with reduced computational and hardware requirements[37]. Specifically, A-mode ultrasound uses single-element transducers (SETs) to generate 1D scans instead of 2D images of the muscle, thereby eliminating the need for computationally intensive image processing algorithms and associated instrumentation. While A-mode ultrasound has been used to classify hand gestures[38–41], muscle states[42], and ambulation modes[43], continuous estimation of muscle mechanical loads is less explored. Only a few studies have investigated estimating hand grip force[44,45] and knee torque[46] during static isometric contractions or elbow torque during controlled isokinetic elbow flexions[47]. The use of A-mode ultrasound for reliable muscle deformation measurement and joint torque estimation during dynamic and unconstrained functional tasks remains unexplored. Hence, there remains a gap to investigate the applicability of A-mode ultrasound for continuous tracking of muscle mechanical loads during unconstrained dynamic activities in real-world environments.

In addition to unconstrained dynamic activities, it is also important to study the applicability of A-mode ultrasound on muscles with different architectures. Skeletal muscles are typically categorized as parallel or pennate based on the orientation of their fascicles[48]. During contraction, a parallel muscle uniformly bulges across its cross-section[21], suggesting a direct relationship between the produced load and changes in thickness or width, which can be captured by A-mode ultrasound. In contrast, a pennate muscle undergoes more complex morphology during contraction, exhibiting a diverse range of cross-sectional shape changes related to both contraction intensity and joint angle[49–51]. As a result, multiple levels of mechanical loads can be observed at the same muscle thickness or width[50,51]. Hence, the effectiveness of using A-mode ultrasound to track the load produced by pennate muscles remains unexplored.

In this work, we aim to develop an approach for joint torque estimation using A-mode ultrasound for both parallel and pennate muscles during unconstrained dynamic activities. To achieve this goal, we first present a method to track muscle thickness with A-mode ultrasound during dynamic motion. Next, we evaluate estimating joint torque using muscle thickness and joint angle during isokinetic contractions of a parallel muscle, the biceps brachii (BB), and a pennate muscle, the rectus femoris (RF). Finally, we demonstrate the feasibility of joint torque estimation using A-mode ultrasound and wearable measures of joint kinematics during a range of unconstrained real-world tasks, including weightlifting, cycling, and both treadmill and outdoor locomotion.

## Results
### Muscle thickness tracking during motion
To enable joint torque estimation, we designed a transducer mount capable of holding four SETs at different angles (Fig. 1a), which can be worn on the limb above the target muscle (Fig. 1b, c). This sensor redundancy was implemented to account for the difference in muscle shape among participants. Specifically, ultrasound echo intensity is highly dependent on the angle of incidence, with the intensity maximized when the ultrasound beam hits the target boundary at a 90° angle (Supplementary Fig. S2). With multiple transducers at a range of angles, we increased the likelihood of normal incidence while collecting from muscles with different sizes and shapes.

To measure muscle thickness, we developed a custom muscle boundary tracking algorithm (MBTA) that identifies the transducer with the strongest echoes and uses its signal to measure muscle boundary depth (Fig. 1d). The MBTA can continuously track thickness changes in both superficial and deep muscles during contractions (Supplementary Fig. S1). In practice, muscle deformation constantly alters the angle between the muscle boundary and the transducer, especially during dynamic motion. To address the resultant variations in ultrasound echo intensity, the MBTA fuses results from two different tracking methods to provide robust estimation of muscle thickness (Methods). To test the MBTA's tracking performance and robustness, we cyclically displaced a SET relative to 3D printed surfaces on a mechanical testing system (Supplementary Fig. S2) and artificially altered the echo intensity to simulate the effect of changing angle of incidence (Supplementary Fig. S3). In reference to the ground truth displacement, the MBTA produced robust tracking with a root mean squared error (RMSE) of 0.05 mm and a normalized root mean squared error (NRMSE) of 0.5%.

Lastly, we conducted an in vivo validation by simultaneously collecting B-mode and A-mode ultrasound while a participant performed isokinetic concentric and eccentric knee extensions on a dynamometer (Supplementary Fig. S4). Compared to the muscle thickness measured with B-mode, A-mode ultrasound achieved an

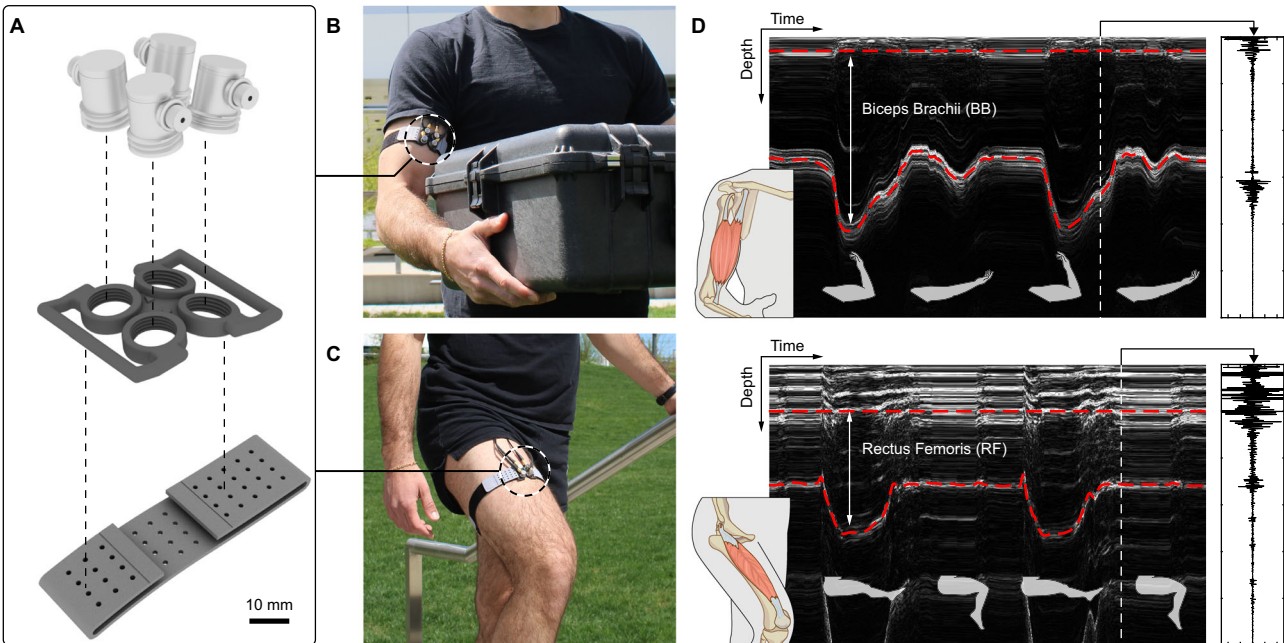

**Fig. 1 | Overview of the A-mode ultrasound system. A** Exploded view of the wearable transducer mount. Four SETs are mounted on a 3D-printed case and attached to the body using a compressive fabric band. **B** Photograph of SETs worn on the upper arm over the BB muscle belly. **C** Photograph of SETs worn on the upper leg over the RF muscle belly. **D** Representative A-mode ultrasound data during concentric contractions of the parallel BB muscle (top) and the pennate RF muscle (bottom). Red dashed lines indicate superficial and deep muscle boundaries measured with the MBTA. Right insets show representative raw ultrasound data at specific time frames.

RMSE of 0.48 mm and a NRMSE of 1.7%, which was smaller than the average thickness of the deep thigh fascia[52].

### Elbow torque estimation during isokinetic contractions

To evaluate the estimation of corresponding joint torque from parallel muscles, we acquired A-mode ultrasound of the BB and elbow kinematics and kinetics with an isokinetic dynamometer on ten healthy adults while performing passive, concentric, and eccentric BB contractions at various speeds (Figs. 1b, 2a, Supplementary Fig. S5). Data from a representative participant showed that the BB thickness exhibited minimal change during passive motion compared to active contractions and varied synchronously with elbow torque in all conditions (Fig. 2b). For each participant, we applied a quadratic fit to data from all conditions (Methods). The quadratic fit, denoted as (MT, Ang)$^2$, uses two input variables, BB thickness and elbow angle, and estimates elbow torque as the target variable (Fig. 2c). Across all participants, the individualized models achieved RMSEs of 3.89 ± 0.86 Nm (mean ± SD), NRMSEs of 7.6 ± 1.4%, and coefficients of determination ($R^2$) of 0.92 ± 0.03 (Supplementary Table S1).

We observed no significant effect of joint velocity on the RMSEs of individualized models ($F_{2,18} = 0.163$, $p = 0.851$) (Fig. 2d). A significant effect of contraction type on the RMSEs was observed ($F_{2,18} = 35.0$, $p < 0.001$) with multiple pairwise comparisons revealed no significant difference in estimation errors between concentric and eccentric contractions ($p = 0.527$) and significantly lower errors during passive motion than those during active contractions (concentric: $p < 0.001$; eccentric: $p < 0.001$) (Fig. 2e). These lower RMSEs aligned with expectations given the minimal torque variation during passive motion. Furthermore, we studied the contributions of input variables by comparing the performance of (MT, Ang)$^2$ to quadratic fits based solely on muscle thickness (MT$^2$) and solely on joint angle (Ang$^2$) (Supplementary Figs. S6a, S6b). We observed significant effects of input variables on estimation accuracy (NRMSEs: $\chi^2_2 = 20$, $p < 0.001$; $R^2$: $F_{2,18} = 1001$, $p < 0.001$). MT$^2$ produced a small and non-significant decline in performance compared to (MT, Ang)$^2$ (NRMSE: $p = 0.312$; $R^2$:

$p = 0.090$), with NRMSEs increasing to 9.6 ± 2.9% and $R^2$ decreasing to 0.87 ± 0.05. In contrast, Ang$^2$ yielded significantly poorer estimations than (MT, Ang)$^2$ (NRMSE: $p < 0.001$; $R^2$: $p < 0.001$), with NRMSEs of 25.8 ± 3.0% and $R^2$ of 0.10 ± 0.05. This model comparison suggested that muscle thickness played a much more substantial role than joint angle for estimating the corresponding joint torque from a parallel muscle. Lastly, we evaluated a generalized model by fitting (MT, Ang)$^2$ on data from all participants. In comparison to individualized models, the generalized model generated an increased NRMSE of 11.5% and a decreased $R^2$ of 0.82 (Supplementary Table S1).

### Elbow torque estimation during dumbbell curls

To investigate functional applications of elbow torque estimation, we performed a study on five participants during dumbbell curls. The study consisted of dynamometer calibration and free weight demonstration. We collected BB thickness with A-mode ultrasound and elbow angle with inertial measurement units (IMUs). During the calibration, participants performed single-speed BB contractions on the dynamometer (Fig. 2a). Using the calibration data, we fitted individualized (MT, Ang)$^2$ models which produced RMSEs of 2.62 ± 0.78 Nm, NRMSEs of 6.8 ± 1.4% and $R^2$ of 0.96 ± 0.02 across all participants (Supplementary Table S2). These models were then used to estimate elbow torque during unconstrained dumbbell curls.

During the demonstration, participants performed dumbbell curls with various free weights (Fig. 3a). Data from a representative participant showed that BB thickness increased with weight, while elbow range of motion decreased (Fig. 3b). We obtained the elbow torque estimation (Fig. 3c) by applying the (MT, Ang)$^2$ model from calibration and compared the results with those calculated from a rigid body model (Fig. 3d, Supplementary Fig. S7, Methods). For all participants, there was a close agreement between the estimated and calculated torque (Supplementary Fig. S8), with the average elbow torque increasing proportionally to the weight of the dumbbell (Fig. 3e, f). Quantitatively, absolute errors in average torque measurements were 1.25 ± 0.71 Nm for no weight, 1.18 ± 0.57 Nm for medium weights, and

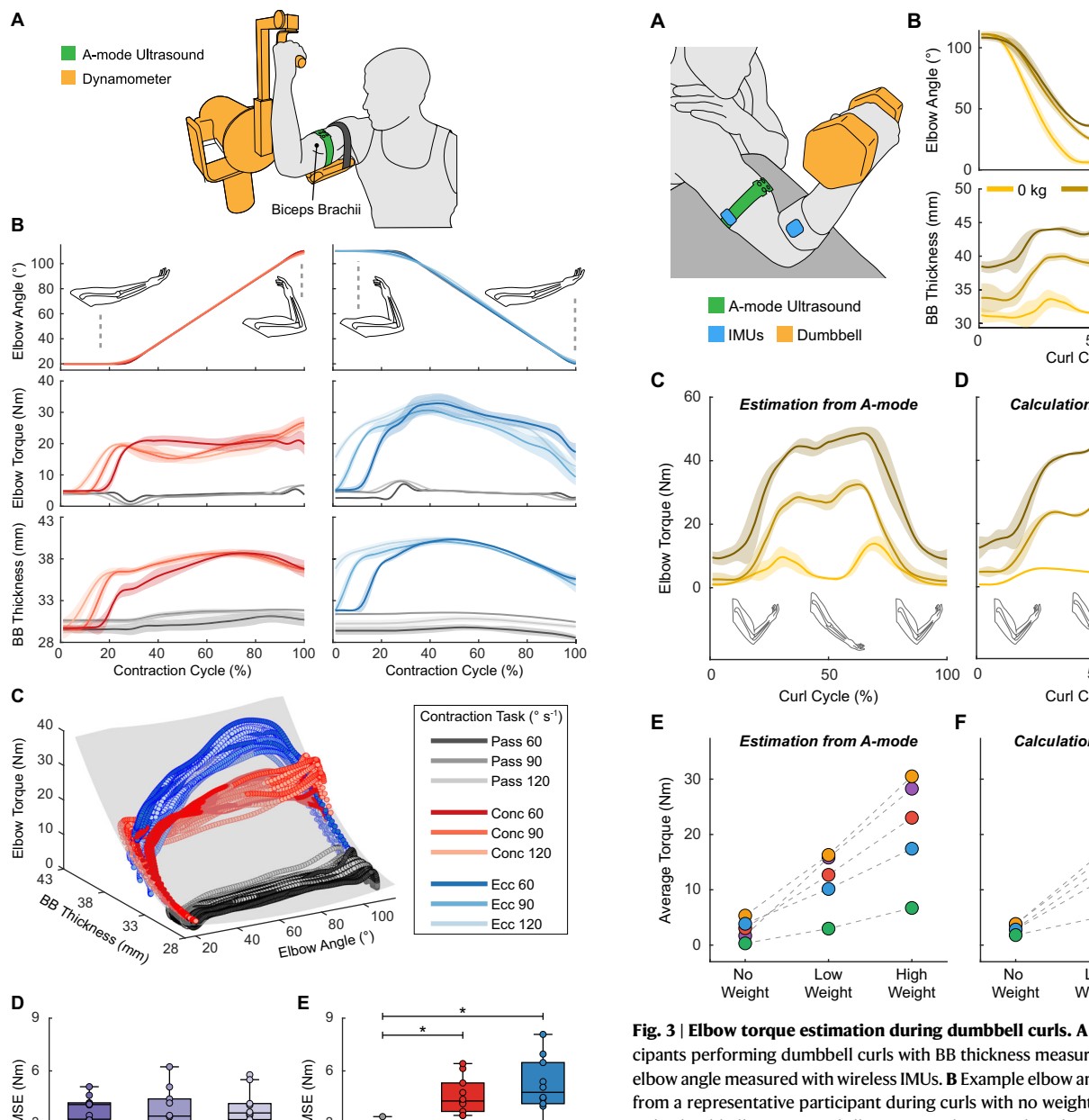

**Fig. 2 | Elbow torque estimation during isokinetic contractions of the biceps brachii. A** Illustration of participants secured on a dynamometer with SETs placed over the BB muscle belly. **B** Example elbow angle, elbow torque, and BB thickness from a representative participant during passive (pass; gray), concentric (conc; red), and eccentric (ecc; blue) contractions at 60° s⁻¹, 90° s⁻¹, and 120° s⁻¹. Lines and shaded regions represent mean ± SD ($n = 7$ contractions). **C** Example relationship between BB thickness, elbow angle, and elbow torque from a representative participant across all conditions, with an overlaid quadratic fit (gray). **D** RMSEs for different contraction speeds across all participants ($n = 10$). One-way ANOVA: no significant main effect ($p = 0.851$, $F_{2,18} = 0.163$). **E** RMSEs for different contraction types across all participants ($n = 10$). One-way ANOVA: significant main effects ($p < 0.001$, $F_{2,18} = 35.0$). Bonferroni post-hoc analysis: pass vs conc ($p < 0.001$), pass vs ecc ($p < 0.001$), and conc vs ecc ($p = 0.527$). For **D** and **E**, each box bounds the interquartile range (IQR) divided by the median with whiskers extending up to 1.5*IQR. Each dot represents the RMSE for one participant. *$p < 0.05$.

**Fig. 3 | Elbow torque estimation during dumbbell curls. A** Illustration of participants performing dumbbell curls with BB thickness measured with SETs and elbow angle measured with wireless IMUs. **B** Example elbow angle and BB thickness from a representative participant during curls with no weight (0 kg), a 5 kg, and a 10 kg dumbbell. **C** Estimated elbow torque during curls with various weights for this participant. **D** Simulated elbow torque for this participant calculated using a rigid body model based on classical mechanics. For **B**–**D**, lines and shaded regions represent mean ± SD ($n = 6$ repetitions). **E** Average estimated torque for all participants ($n = 5$) at different weight conditions. **F** Average calculated torque for all participants ($n = 5$) at different weight conditions. For **E** and **F**, each color represents data for one participant. Each dot represents the average torque of all contractions ($n = 6$) within the respective condition.

2.41 ± 0.94 Nm for heavy weights across all participants. Qualitatively, we observed similar double-peak torque profiles from both models, with peak locations aligned with instances when the forearm was perpendicular to gravity and the moment arm was maximized (Fig. 3c, d). Additionally, the first peaks had lower amplitudes, as they occurred during elbow extension when gravity aligned with the motion, while the second peaks occurred during elbow flexion when gravity needed to be counteracted. We further noted that our estimation tends to produce higher torque values at these peak locations compared to the rigid body calculations (Supplementary Fig. S8). Such discrepancy is likely attributed to factors such as muscle co-contraction, which the rigid model cannot capture, as well as the various simplifying assumptions employed in the rigid body model (Methods).

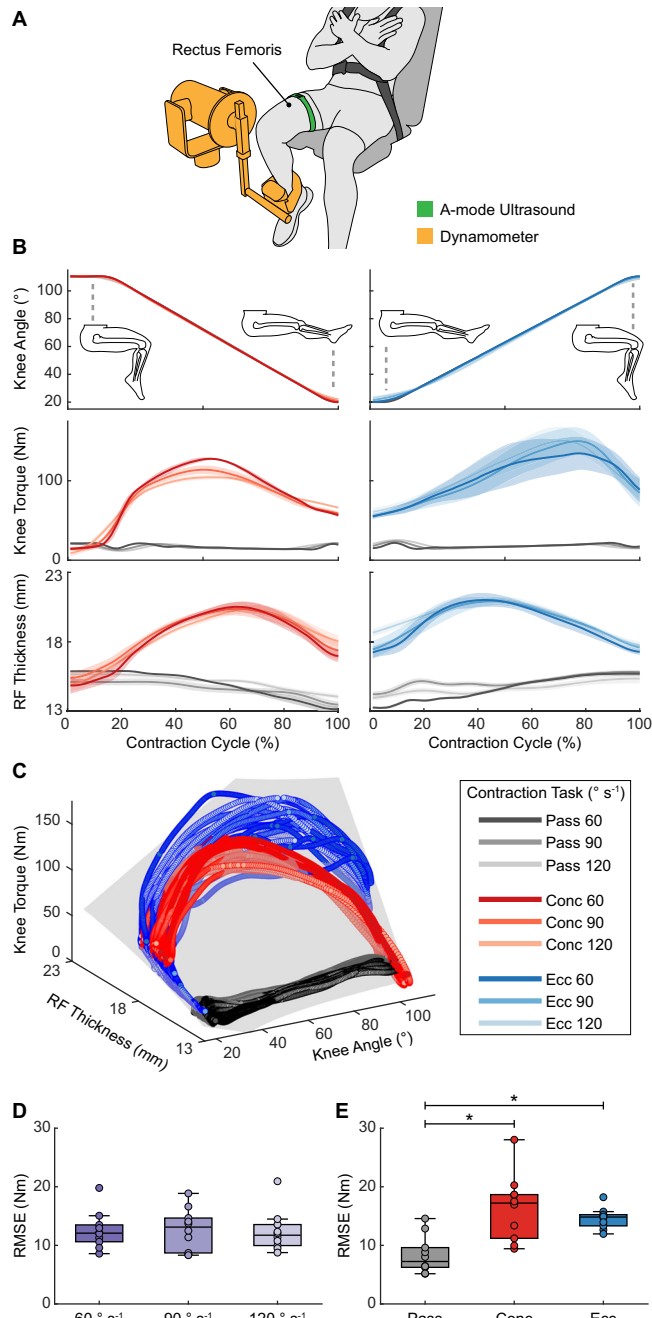

## Fig. 4 | Knee torque estimation during isokinetic contractions of the rectus femoris.

**A** Illustration of participants secured on a dynamometer with SETs placed over the RF muscle belly. **B** Example knee angle, knee torque, and RF thickness from a representative participant during passive (pass; gray), concentric (conc; red), and eccentric (ecc; blue) contractions at 60° s⁻¹, 90° s⁻¹, and 120° s⁻¹. Lines and shaded regions represent mean ± SD ($n = 7$ contractions). **C** Example relationship between RF thickness, knee angle, and knee torque from a representative participant across all conditions, with an overlaid quadratic fit (gray). **D** RMSEs for different contraction speeds across all participants ($n = 10$). Friedman's test: no significant main effects ($p = 1.00$, $\chi^2_2 = 0.000$). **E** RMSEs for different contraction types across all participants ($n = 10$). One-way ANOVA: significant main effects ($p < 0.001$, $F_{2,18} = 21.8$). Bonferroni post-hoc analysis: pass vs conc ($p < 0.001$), pass vs ecc ($p < 0.001$), and conc vs ecc ($p = 0.894$). For **D** and **E**, each box bounds the IQR divided by the median with whiskers extending up to 1.5*IQR. Each dot represents the RMSE for one participant. *$p < 0.05$.

## Knee torque estimation during isokinetic contractions

To evaluate the estimation of corresponding joint torque from pennate muscles, we collected A-mode ultrasound of the RF and knee kinematics and kinetics with the dynamometer on ten participants during isokinetic RF contractions (Figs. 1c, 4a, Supplementary Fig. S5). Data from a representative participant showed a correlated yet non-monotonic relationship between RF thickness and knee torque, as evidenced by the mismatch between peak locations of RF thickness and knee torque during active contractions (Fig. 4b). By fitting (MT, Ang)² on data from each participant across all contractions (Fig. 4c), we obtained individualized models with RMSEs of 12.59 ± 3.12 Nm, NRMSEs of 7.0 ± 1.3%, and $R^2$ of 0.92 ± 0.02 (Supplementary Table S1).

Joint velocity showed no significant effect on the RMSEs ($\chi^2_2 = 0.000$, $p = 1.00$) (Fig. 4d), whereas contraction type showed a significant effect ($F_{2,18} = 21.8$, $p < 0.001$). Despite the significantly lower RMSEs during passive motion than active contractions (concentric: $p < 0.001$; eccentric: $p < 0.001$), there was no significant performance difference between concentric and eccentric contractions ($p = 0.894$) (Fig. 4e). Furthermore, during input contribution analysis, quadratic models with different inputs showed significantly different performances (NRMSE: $F_{2,18}) = 275$, $p < 0.001$; $R^2$: $F_{2,18} = 434$, $p < 0.001$) (Supplementary Figs. S6c, S6d). Both MT² and Ang² models produced significantly poorer fit than the (MT, Ang)² model (NRMSE of MT²: $p < 0.001$; $R^2$ of MT²: $p = 0.003$; NRMSE of Ang²: $p < 0.001$; $R^2$ of Ang²: $p < 0.001$), with Ang² yielding the worst fit (NRMSEs = 23.6 ± 1.9%, $R^2 = 0.16 ± 0.06$), followed by MT² (NRMSEs = 11.6 ± 3.4%, $R^2 = 0.79 ± 0.11$). Such drastic performance difference suggested that both muscle thickness and joint angle were important for estimating the corresponding joint torque from a pennate muscle, with muscle thickness playing a larger role than joint angle. By fitting (MT, Ang)² on data from all participants, the generalized model produced an increased NRMSE of 12.0% and a decreased $R^2$ of 0.78 (Supplementary Table S1).

Lastly, we performed a single-participant sensitivity analysis to understand the effect of transducer placement (Supplementary Text). By collecting A-mode ultrasound at 18 different locations across the RF, we observed that transducer placement largely affected the ultrasound quality of the deep RF boundary but did not dramatically affect the correlation to knee torque (Supplementary Fig. S9). However, applying the fit obtained from a specific location to neighboring locations led to increasing estimation errors, highlighting the importance of minimizing sensor placement drift during use.

## Knee torque estimation during cycling

To investigate functional applications, we performed a single-participant study to estimate knee torque during stationary cycling at varying resistance levels. To achieve knee torque estimation, we collected A-mode ultrasound of the RF and vastus lateralis (VL) muscles and captured knee angle using IMUs (Supplementary Text). Qualitatively, estimated knee torque from both the RF and VL increased with pedaling resistance (Supplementary Fig. S10).

## Knee torque estimation during treadmill and outdoor locomotion

To evaluate knee torque estimation during unconstrained dynamic activities, we performed a treadmill study on five participants and a single-participant outdoor locomotion demonstration. We collected RF thickness using A-mode ultrasound and joint kinematics with IMUs. We primarily focused on the load-bearing stance phase of gait, segmented using IMU signals (Methods).

During treadmill locomotion, we obtained ground truth knee torque using inverse dynamics while participants walked and ran at varying speeds and slopes on an instrumented treadmill (Fig. 5a). By fitting quadratic fits on data from each participant across all treadmill conditions, we obtained individualized models with RMSEs of

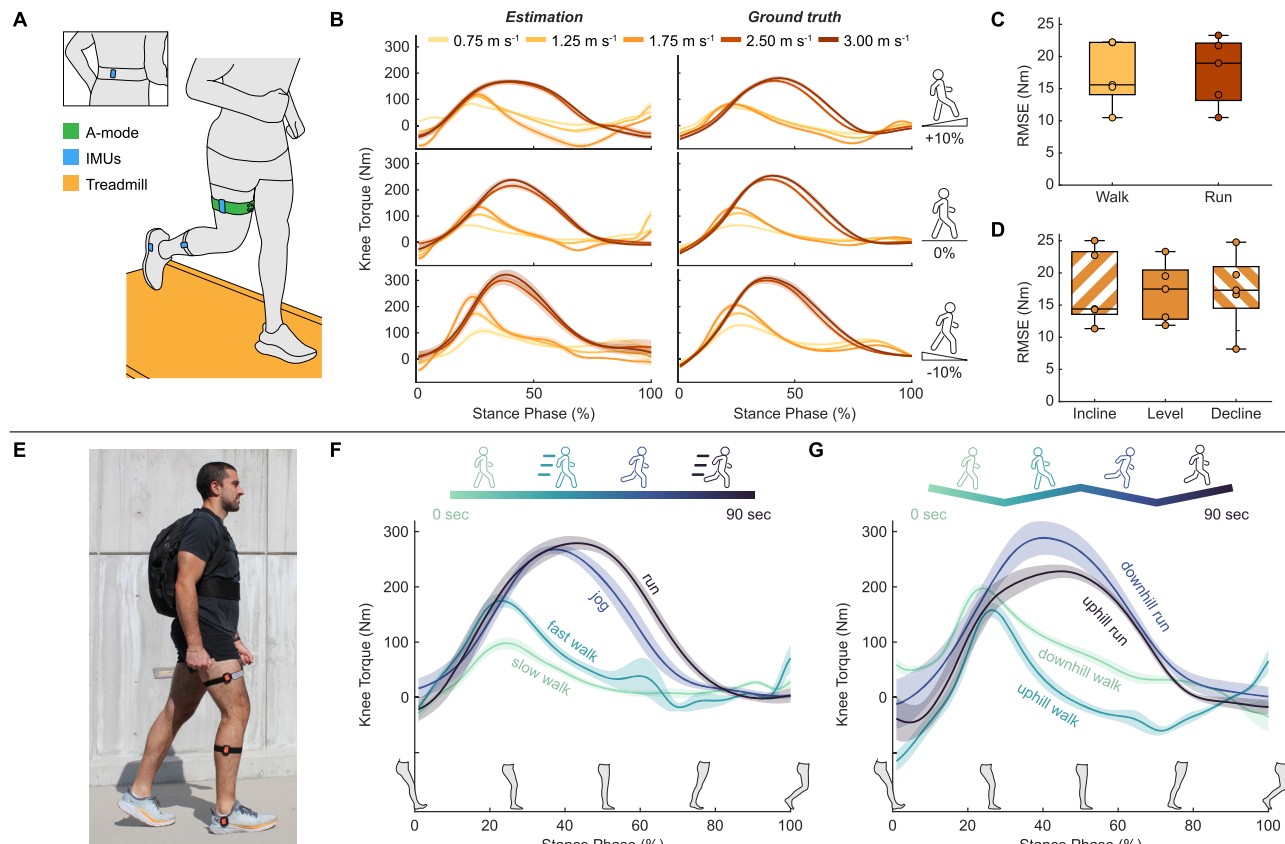

**Fig. 5 | Knee torque estimation during treadmill and outdoor locomotion.**
**A** Illustration of participants performing various walking and running tasks on an instrumented treadmill with RF thickness measured with SETs and joint kinematics measured with wireless IMUs. **B** Example estimated knee torque during the stance phase (left column) and corresponding ground truth (right column) for treadmill locomotion at various speeds (0.75, 1.25, 1.75 m s⁻¹ walking and 2.50, 3.00 m s⁻¹ running) and slopes [+10% (top row), 0% (middle row), −10% (bottom row)] from a representative participant. Lines and shaded regions represent mean ± SD across the steps within each condition. **C** RMSEs for different locomotion types across all participants ($n = 5$). Student's two-tailed $t$ tests: walk vs run ($p = 0.873$). **D** RMSEs for

different slope levels across all participants ($n = 5$). One-way ANOVA: no significant main effect ($p = 0.965$, $F_{2,8} = 0.036$). For **C** and **D**, each box bounds the IQR divided by the median with whiskers extending up to 1.5*IQR. Each dot represents the RMSE for one participant. **E** Photograph of a participant performing outdoor locomotion with a backpack containing the A-mode ultrasound electronics. Placements of SETs and IMUs remained unchanged compared to the treadmill test. **F** Knee torque estimation at different speeds during level ground outdoor locomotion. Lines and shaded regions represent mean ± SD ($n = 15$ steps). **G** Knee torque estimation during comfortable-speed downhill and uphill walking and running. Lines and shaded regions represent mean ± SD ($n = 15$ steps).

$17.43 \pm 5.09$ Nm, NRMSEs of $6.0 \pm 1.1\%$, and $R^2$ of $0.92 \pm 0.03$ (Supplementary Table S2). Data from a representative participant is illustrated to show the correlation between the estimated and the ground truth knee torque (Fig. 5b). Across participants, we observed no significant difference in RMSEs during walking and running ($p = 0.873$) (Fig. 5c). Treadmill slope also had no significant effect on RMSEs ($F_{2,8} = 0.036$, $p = 0.965$) (Fig. 5d). These results were generated using quadratic models with RF thickness, along with IMU-measured pelvis, thigh, and shank angles as input variables. These input variables were used to capture the combined effect of muscle deformation and joint kinematics on knee torque estimation.

As proof of concept for real-world use, one participant further performed an outdoor locomotion demonstration by wearing a backpack containing the data collection electronics (Fig. 5e). The demonstration consisted of two parts: (1) walking and running at various speeds on level ground and (2) walking and running at a self-selected comfortable speed on slopes. Applying the individualized model obtained from the treadmill test, we estimated an increase in knee torque with increasing speeds during the outdoor conditions, with peak knee torque increased by 78.4% from slow to fast walking and by 4.3% from jogging to running (Fig. 5f). These percentage increases aligned with those observed in the treadmill test, where the peak knee torque for this participant increased by 71.3% from 0.75 m s⁻¹

to 1.25 m s⁻¹ walking and by 5.7% from 2.50 m s⁻¹ to 3.00 m s⁻¹ running (Fig. 5b). Additionally, we observed a decrease in knee torque with increasing slope, with peak knee torque decreasing by 20.0% from downhill to uphill walking and by 20.9% from downhill to uphill running (Fig. 5g). These percentage changes were smaller than those observed in the treadmill test when comparing the decline and incline conditions (42.9% decrease at 0.75 m s⁻¹ and 46.7% decrease at 2.50 m s⁻¹) (Fig. 5b). This difference was likely due to the smaller slope change in the outdoor test (−7.5% to 7.5%) compared to the treadmill test (−10% to 10%)[53].

## Discussion

In this work, we demonstrated the use of A-mode ultrasound to track muscle thickness during dynamic movements. Subsequently, we showed that a quadratic fit with muscle thickness and joint angle enabled joint torque estimation in various dynamic tasks, ranging from controlled isokinetic contractions to dynamic functional applications. Additionally, we found that muscle thickness played a substantial role in joint torque estimation for both parallel and pennate muscles, while the contribution of joint angle varied based on muscle architecture.

Our joint torque estimation achieved results comparable to those in published studies during controlled dynamic contractions. During

the dynamometer tests, we obtained elbow and knee torque estimation with less than 7.6% NRMSEs and $R^2$ values greater than 0.92 during isokinetic BB and RF contractions at various speeds (Supplementary Table S1). Prior studies on controlled contractions have reported elbow torque estimation with an NRMSE of 9.6% during isokinetic BB contractions by combining EMG with A-mode ultrasound[47] and knee torque estimation with an NRMSE of 15% during isokinetic knee extension using soft strain sensors[25]. However, the applicability of these existing methods for estimating joint torque during unconstrained dynamic tasks remains unexplored.

We further showed that our estimation models could be used in dynamic functional activities in unconstrained environments. We demonstrated joint torque estimation during dumbbell curls and outdoor locomotion with a fully wearable setup including wireless IMUs, A-mode ultrasound, and electronics housed in a backpack. During dumbbell curls, our estimated elbow torques increased with heavier weights and exhibited time series profiles resembling those from a rigid body model (Fig. 3, Supplementary Fig. S8). Given the essential role of the elbow in various throwing motions such as baseball pitch, basketball pass, and javelin throw, continuous monitoring of elbow torque during sports activities holds the potential to facilitate more effective athletic training programs and targeted injury prevention strategies[3,8,54,55]. During outdoor locomotion, the estimated knee torque increased with speed and decreased with incline, with much higher amplitudes during running than walking (Fig. 5). These torque profiles are consistent with previously reported values measured with lab-based equipments[53,56]. Knee disorders can largely affect people's mobility and quality of life. Monitoring knee torque during unconstrained locomotion in communities can be beneficial for preventing and managing conditions like knee osteoarthritis[57] or assessing recovery progress after an ACL reconstruction surgery[58]. Moreover, given the prevalence of work-related lateral epicondylitis (tennis elbow)[59] and occupational knee disorders[60], tracking elbow and knee torque while conducting physically demanding jobs could also help facilitating better workplace ergonomics and preventing work-related injuries.

In addition to its potential use in dynamic applications, our A-mode-based torque estimation has the advantage of providing muscle-specific measurements. In the cycling test, we observed increased torque estimations with pedaling resistance, but with different profiles for RF and VL-based estimations (Supplementary Fig. S10). These estimation profiles align with previously reported EMG activation patterns of the RF and VL[61,62]. Specifically, activation/torque estimates from the VL peak during the downstroke phase, whereas those from the RF peak during the setup phase, which immediately precedes the downstroke phase. This difference in estimated torque is likely due to variations in muscle function, as the RF is responsible for both knee extension and hip flexion, while the VL solely functions as a knee extensor[26]. This demonstrated muscle specificity distinguishes our ultrasound-based approach from other surface level measurements like shear wave tensiometers, EMG, MMG, FMG, and soft strain sensors, demonstrating its potential as a valuable tool for detailed studies of muscle function and biomechanics[16].

Driven by A-mode ultrasound's muscle-specific measurements, we identified that contributions of muscle thickness and joint angle to joint torque estimation were closely associated with the underlying muscle architectures. We found that muscle thickness alone sufficed to capture the overall deformation of a parallel muscle (e.g., the BB) and achieved good correlation with corresponding joint torque (Supplementary Figs. S6a, S6b). This finding aligns with the deformation behavior of parallel muscles, which thicken and widen during contraction due to fascicle shortening[21]. In contrast, for a pennate muscle (e.g., the RF), muscle thickness alone was insufficient for accurate joint torque estimation (Supplementary Figs. S6c, S6d), consistent with prior findings using B-mode ultrasound[50,63]. The

insufficiency of muscle thickness alone occurs because pennate muscles undergo both fascicle shortening and rotation, resulting in a wide range of possible changes in thickness and width[21,50,51]. To capture the combined effect of changes in fascicle length and pennation angle, we showed that both RF thickness and joint angle were necessary for accurate knee torque estimation.

To describe the relationship of muscle thickness and joint kinematics to joint torque, we used a simple $2^{nd}$ order polynomial fit, which offers advantages in model generalizability and transferability due to its simple architecture[64,65]. Specifically, we showed in the dynamometer tests that applying the quadratic fit to all contractions yielded comparable results across different contraction speeds (Figs. 2d and 4d) and types (concentric vs eccentric) (Figs. 2e and 4e). We further showed in the dumbbell curl and the outdoor locomotion tests that quadratic models trained using controlled contractions can generate promising results in unconstrained dynamic tasks (Figs. 3 and 5, Supplementary Fig. S8). In contrast, prior A-mode ultrasound studies have primarily used more advanced machine learning algorithms, such as deep neural networks, by training them directly on the raw ultrasound data. Despite enabling various classification applications[39,41–43] and continuous joint kinematics[66] or static grip force[44,45] estimation, these algorithms often require substantial amounts of training data due to the large number of tunable parameters within the model. In comparison, we alleviated the data requirement by training the relatively simple quadratic fit on the extracted muscle thickness, rather than on the raw A-mode data. As an example, we trained the elbow torque estimator for dumbbell curls using only 15 isokinetic contractions (5 passive, 5 concentric, 5 eccentric). In the future, we expect that combining a feature extraction method, like the MBTA, with advanced machine learning algorithms could enable more accurate torque estimation, potentially by fusing multiple measurements from the same muscle or leveraging measurements from different muscles.

While A-mode ultrasound showed promise for joint torque estimation during unconstrained dynamic activities, we note a few limitations that future work should investigate. First, we used joint torque, instead of muscle force, as ground truth. Given our muscle-level measurements, we could expect improved performance with muscle force as ground truth. However, forces of specific muscles have been infeasible to directly measure non-invasively[8] and require complex computational models to simulate[16,67]. Second, our current fit identification was conducted on an individual basis that required lab-based equipment, such as a dynamometer or an instrumented treadmill. Despite the promise of our generalized models (with less than 5% additional NRMSEs compared to individualized models) (Supplementary Table S1), developing a more accessible calibration procedure, such as one based on body weight, could enhance practicality and improve estimation accuracy for daily use. Notably, such a calibration procedure is a common need for non-invasive wearable sensors, including the shear wave tensiometer[13], soft strain sensors[25,68], and surface EMGs[19,20]. Third, while the MBTA demonstrates potential for real-time implementation (Methods), current torque estimation is conducted in post-processing. Future efforts could focus on developing a centralized system to simultaneously log ultrasound and IMU data and perform the torque estimation algorithm in real-time. Lastly, our study focused primarily on superficial muscles (e.g., BB and RF) and hinge joints (e.g., elbow and knee). While the MBTA can track the thickness of deep muscles (Supplementary Fig. S1), using deep muscle dynamics for estimating joint torque, especially in more complex ball-and-socket joints, remains interesting and unexplored. Despite these limitations, our study presents a step towards using wearable A-mode ultrasound for monitoring muscle and joint mechanical loads in various real-world tasks and environments.

## Methods

### Amplitude-mode ultrasound system

Four 5 MHz SETs (Alpha 113-124-660, Waygate Technologies, Germany) were mounted in a custom 3D printed case (35 mm by 43 mm) at angles of 0°, 1.25°, 2.5°, and 3.75°. The choice of four transducers was based on preliminary tests as this number provided a good balance between capturing signals with sufficient echo strength and achieving reasonable technical specifications including assembly size and frame rate (90 Hz). These transducers were worn on the target limb using a compression band (Scosche, USA) (Fig. 1a). Thin layers of ultrasound gel (Aquasonic, USA) were applied to SETs before donning. The transducer assembly had a total mass of 35.1 g (excluding the transducer cables). Raw ultrasound data of the four SETs were collected with an ultrasonic pulser/receiver (OPBOX, Optel, Poland) and a multiplexer (OPMUX, Optel, Poland) using the pulse echo mode. The ultrasonic pulser operated as a short-circuit step pulser with pulse voltage, pulse charging time, and pulse repetition frequency set to 240 V, 3.1 μs, and 360 Hz (for 4 SETs), respectively. The receiver recorded echoes up to a depth of 80 mm with amplification gain, analog band-pass filter, and sampling frequency set to 20 dB, 4–6 MHz, and 50 MHz, respectively. 1540 m s$^{-1}$ was used as the speed of sound in human tissues. For system wearability, we powered the ultrasound instrument with a rechargeable battery pack (RRC2054, RRC Power Solutions, Germany) and housed all electronics in a backpack (Fig. 5e). The backpack had a total mass of 5.4 kg.

### Muscle boundary tracking algorithm (MBTA)

We developed a custom MATLAB (MathWorks, USA) algorithm to track the time-series depth change of the target muscle boundary (Fig. 1d, Supplementary Fig. S1). We first manually cropped the data from each SET to the depth range containing only the muscle boundary of interest (Supplementary Fig. S1b). The MBTA then identified the SET with the optimal angle of incidence by averaging the peak echo intensity across all time frames and selected the transducer (among the 4 SETs) with the largest mean peak echo intensity. Once identified for a given muscle, data from the same SET was used across all experimental conditions.

Next, the cropped data from the selected SET were fed into a brightness-based and a cross-correlation-based tracking method, which were later fused to obtain the final tracking. The brightness-based method performed a Hilbert transform on the raw ultrasound data for envelope detection, applied a 2D box filter (2.5 mm by 0.05 sec) for smoothing, and tracked the peak location of the smoothed signal at each time frame. The brightness-based method tracked the general movement of the muscle boundary but was susceptible to variations in ultrasound echo intensity, resulting in high-frequency noise in the tracking (Supplementary Fig. S1c). These echo intensity fluctuations may be from motion induced soft tissue artifacts or muscle fascicle rotation during contraction. The cross-correlation-based method applied cross-correlation to the raw ultrasound data of consecutive time frames and integrated the relative delays across all time frames. The cross-correlation-based method produced smooth tracking, but with low frequency drift (Supplementary Fig. S1d). The drift resulted from errors accumulated during numerical integration. Finally, the MBTA summed the low-pass filtered (1.5 Hz) tracking output from the brightness-based method with the high-pass filtered (1.5 Hz) tracking output from the cross-correlation-based method. Combining the two methods resulted in depth tracking that was robust against high-frequency noises and the low-frequency drift (Supplementary Figs. S1e, S1f).

Although the MBTA is currently implemented in post-processing, its required computational steps are straightforward. The algorithm operates with an average runtime of less than 0.3 milliseconds per frame on a standard CPU (Intel Core i7-10750H @ 2.60 GHz), showcasing its potential for real-time use. To transition to real-time implementation, further developments could involve pre-defining the region of interest based on body measures, such as weight and height, replacing low-pass filters with the simpler moving average filters, and implementing the algorithm on a field programmable gate array (FPGA). Notably, the 2D box filter in the brightness-based method and the low-pass and high-pass filters in the final fusion step may introduce time delays in tracking. Therefore, it is important to optimize filter parameters, like filter size and cutoff frequency, for specific applications to minimize these delays.

Lastly, muscle thickness was calculated by subtracting the depth of the superficial muscle boundary from that of the deep boundary. For each participant, we used the MBTA to measure the deep boundary depth, while assuming the superficial boundary depth as a manually measured constant. This assumption was used because both studied muscles (BB and RF) were superficial muscles, which exhibited minimal depth changes in their superficial boundaries during contractions (Supplementary Fig. S1a).

### Experimental methods and data processing

**Participants.** Ten healthy adults (3 females and 7 males; age = 29.0 ± 3.7 years; mass = 70.6 ± 12.2 kg; height = 1.75 ± 0.10 m; Supplementary Table S3) participated in the dynamometer test. Five healthy adults (1 female and 4 males; age = 27.8 ± 2.4 years; mass = 76.8 ± 12.0 kg; height = 1.79 ± 0.12 m; Supplementary Table S4) participated in the dumbbell curl and treadmill locomotion tests. One healthy male (age = 29 years; mass = 83 kg; height = 1.83 m) participated in the transducer placement, stationary cycling, and outdoor locomotion tests. All participants provided written informed consent before participation. The study was approved by the Harvard Medical School Institutional Review Board (IRB #22086), and all tests were carried out in accordance with the approved study protocol. The authors affirm that the participant appearing in the photograph in Fig. 5 provided written consent for its publication.

**Dynamometer testing.** Ten participants attended two isokinetic dynamometer (HUMAC Norm, CSMi Solutions, USA) tests: an elbow test (Fig. 2a) and a knee test (Fig. 4a). We evaluated the right arm and the right leg for all participants. SETs were placed over the BB and RF muscle belly to capture muscle deformation. For each test, participants performed seven repetitions each of passive, concentric, and eccentric contractions at three speeds (60, 90, 120° s$^{-1}$), totaling 63 repetitions per test (Supplementary Fig. S5). For both tests, the range of motion was set to 20°–110° of elbow/knee flexion, with 0° defined as full extension for both joints. Participants were instructed to fully relax during passive motion and exert near-maximal effort during active contractions. To mitigate fatigue, participants were given 1-minute rest between active contraction trials and were allowed additional rest whenever needed. Joint angles and torques were measured from the dynamometer and collected using an external data acquisition unit (DAQ; 10 kHz; PowerLab 8/35, AD Instruments, New Zealand). Ultrasound sync pulses (360 Hz) were simultaneously recorded by the DAQ. Joint angles and torques were resampled in post-processing to match the frequency of ultrasound data for synchronization.

Joint torque, joint angle, and muscle thickness data were low-pass filtered using a fourth-order, zero-lag, Butterworth filter at 5 Hz[25]. Notably, torques measured by a dynamometer include both the gravitational torque (from the weight of the limb and the dynamometer attachment) and the biological torque from muscle contraction[69]. To isolate biological torque changes, we first modeled gravitational torque (torque measurements during passive contractions) as a sinusoidal function to joint angle and subsequently subtracted this function from torque measurements in all passive and active contractions. We segmented the corrected torque, angle, and thickness data to the torque generation phase of active contractions and to joint flexion and extension phases of passive motion. We then

time-normalized these segmented data to account for variations in the number of data points resulting from different contraction speeds (Figs. 2b and 4b). For the generalized models, each participant's joint torque and muscle thickness data were normalized to their maximum values to account for individual differences in muscle size and strength. We used $R^2$, RMSE, and NRMSE to evaluate the joint torque estimation. Specifically, the NRMSE was calculated by dividing the RMSE by the range of ground truth joint torque.

**Dumbbell curl testing.** Five participants attended the dumbbell curl test. During calibration, the participant performed five repetitions each of passive, concentric, and eccentric BB contractions at 90° s⁻¹ on the dynamometer following the same instructions as in the dynamometer test (Fig. 2a). Subsequently, participants performed six dumbbell curls each with no weight, a self-selected medium weight dumbbell (e.g., 1.5 kg, 3 kg, or 5 kg), and a self-selected heavy weight dumbbell (e.g., 3 kg, 7 kg, or 10 kg) on a bench with 30° incline. The participant wore SETs on the right upper arm over the BB belly and two wireless IMUs (Movella DOT, Movella, USA) on the lateral sides of the right upper arm and forearm (Fig. 3a). Sensor placements were not changed between the two parts of the test. Elbow angles were measured using IMUs and ground truth elbow torque was recorded from the dynamometer during calibration. IMU measurements (120 Hz) and A-mode ultrasound were synchronized post hoc by simultaneously recording an impact event using the SETs and an additional IMU placed directly over the SETs.

We processed the calibration data using the same approach as in the dynamometer test and fitted a $(MT, Ang)^2$ model using the IMU-based elbow angle (see "IMU-based kinematics calculation and gait segmentation"). During dumbbell curl validation, elbow angle and BB thickness data were low-pass filtered at 5 Hz and fed into the $(MT, Ang)^2$ model for elbow torque estimation. The torque estimation was segmented by peak elbow angle, time-normalized, and evaluated by comparing to the calculation from a rigid body model (see "Dumbbell curl rigid body model").

**Locomotion testing.** Five participants attended the treadmill locomotion test. During the test, the participant walked and ran at five speeds (walking: 0.75, 1.25, 1.75 m s⁻¹; running: 2.50, 3.00 m s⁻¹) on three levels of slope (5.71°/10%, 0°, −5.71°/−10%). The participant performed each task for 30 seconds with lower-limb kinematics measured using a motion capture (mocap) system (200 Hz; Qualisys, Sweden) and 3D ground reaction forces (GRF) measured using an instrumented split-belt treadmill (2 kHz; Bertec, USA). The participant wore SETs on the right leg over the RF belly and four wireless IMUs on lateral sides of the right foot, shank, and thigh, as well as on the back of the pelvis (Fig. 5a). A-mode ultrasound was synchronized with the mocap system using the DAQ.

Following the treadmill test, one participant proceeded to the outdoor locomotion test. He kept the same placements of SETs and IMUs, doffed the mocap markers, and donned the backpack (Fig. 5e). The outdoor test contained two experiments. First, the participant performed 15 steps of slow walking, fast walking, jogging, and running on a level outdoor walkway. Then, the participant performed 15 steps of downhill walking, uphill walking, downhill jogging, and uphill jogging on an outdoor ramp inclined at 4.3° or 7.5% [measured using an angle gauge (GemRed, China)]. Each experiment lasted approximately 90 s.

We calculated knee, hip, and thigh angles using IMUs. During the treadmill test, we performed inverse dynamics analysis using mocap and GRF data to calculate ground truth knee torque (Visual3D, C-Motion, USA). For both treadmill and outdoor tests, all data were low-pass filtered at 10 Hz and segmented to the stance phase of gait (see "IMU-based kinematics calculation and gait segmentation"). We obtained the quadratic fit using data from the treadmill test and

applied it to the outdoor test to demonstrate knee torque estimation during outdoor locomotion tasks. We used IMU measured pelvis, thigh, and shank angles as the kinematics inputs to the fitting model. These inputs were used to account for the biarticular nature of the RF muscle, whose geometry is affected by both the hip and knee joints. Notably, hip angles were not considered during the dynamometer tests because participants were secured to the chair with minimal change in their hip angles.

## Quadratic fit on muscle thickness and joint kinematics
The non-linear relationship between muscle deformation and force production has been widely reported in literature[27,48,50,70]. Research in muscle modeling has incorporated muscle geometric models to complement force simulations. However, there remains a trade-off between computational simplicity (e.g., constant thickness or area assumptions) and the complete representation of muscle shape changes (e.g., 3D finite element methods)[16]. Hence, it remains challenging to develop accurate and accessible biomechanical models that comprehensively capture the muscle's complex 3D deformation during contraction, not to mention the interactions with neighboring muscles[16]. In practice, this challenge has led to a rather empirical approach in defining the relationship between muscle deformation measurements and joint torque estimates. Specifically, a spectrum of models, including quadratic[28], cubic[25,70], exponential[71], and even more complex machine learning[44–47] models have been suggested to describe such relationship. In this work, we chose the quadratic fit, mainly for its simplicity and interpretability, for describing the mapping from muscle thickness and joint kinematics to joint torque.

## Dumbbell curl rigid body model
For the dumbbell curl test, we modeled the forearm and hand as a rigid rod with uniformly distributed mass and the dumbbell as a point mass at the distal end of the rod (Supplementary Fig. S7). The torque from elbow flexors, $T_{elbow}$, can be expressed as

$$T_{elbow} = \frac{m_0 g L}{2} \sin\theta + mgL \sin\theta + I\alpha, \tag{1}$$

where $m_0$ is the total mass of the forearm and the hand, $m$ is the mass of the dumbbell, $g$ is the gravitational acceleration, $L$ is the distance from the cubital fossa (elbow pit) to the center of the third metacarpal bone (center of the hand) at 0° wrist flexion, $\theta$ is the angle between the forearm and the line of gravity, $I$ is the moment of inertia, and $\alpha$ is the angular acceleration. $L$ was directly measured from the participants, $m_0$ was calculated using the reported body segment weights[72], $\theta$ was measured using IMUs, and $\alpha$ was calculated by taking the second derivative of $\theta$. The moment of inertia of the forearm and hand was calculated using

$$I = m_0 \left(\frac{L}{2}\right)^2 + mL^2. \tag{2}$$

## IMU-based kinematics calculation and gait segmentation
The IMUs used in this work provided 3D orientation data, which were generated using the manufacturer's proprietary algorithms. However, the susceptibility of IMU orientations to yaw drift is widely documented[73,74]. To correct for drift, we assumed perfect hinge joints for the elbow, knee, and hip. We also assumed our IMU placement guaranteeing that one of its local coordinate axes perfectly aligned with the joint axis[74]. With these assumptions, each IMU can measure the angle of respective body segment (forearm, upper arm, pelvis, thigh, shank, and foot) relative to gravity. Specifically, we measured these segment angles by redefining the IMU global frame (with one axis aligned with the joint axis and another pointed against gravity)

and finding the angle between the local and the updated global frames around the joint axis. Joint angles can then be calculated by subtracting the angles of two adjacent segments (e.g., hip angle = pelvis angle – thigh angle, knee angle = thigh angle – shank angle).

During treadmill/outdoor locomotion tests, the stance phase of gait was segmented and defined as the period between heel-strikes and subsequent toe-offs. Heel-strikes were identified by detecting the timing of local maxima in foot angles, and toe-offs were identified by detecting the timing of local minima in foot velocities that occurred between consecutive heel-strikes.

## Statistics

Statistical analyses were performed to evaluate effects of joint velocity, contraction type, model input, and treadmill locomotion condition (walking vs running, slope level) on joint torque estimation. During these analyses, we first performed the Shapiro–Wilk test to check data normality. For normally distributed data, we used one-way repeated measures analysis of variance (ANOVA) to evaluate the metric's main effect. For non-normally distributed data, we applied Friedman's test for main effect analysis. Once the main effect was found to be significant, post-hoc analyses with Bonferroni correction were conducted for multiple pairwise comparisons (two-tailed). For pairwise comparisons on normally distributed data (such as RMSEs for treadmill walking versus treadmill running), we used the two-tailed two-sample t-test. The level of statistical significance was set at $p < 0.05$.

## Reporting summary

Further information on research design is available in the Nature Portfolio Reporting Summary linked to this article.

## Data availability

All data supporting the findings of this study are available within the article and its supplementary files. Any additional requests for information can be directed to, and will be fulfilled by, the corresponding author. Source data are provided with this paper.

## Code availability

All code for this work will be made available from the corresponding author upon request.

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

## Acknowledgements

We thank Sherry Wang, Michael Lichfield, Vinicius Cene, Prabhat Pathak, Oluwaseun Araromi, Daekyum Kim, Lauren Baker, Xiyuan Huang, the Wyss Institute Clinical Research Team, and our participants for their contributions to this work. This work was supported by National Institutes of Health grant R21AR076686 (R.D.H.), National Science Foundation [award #2236157 (C.J.W.) and DGE2140743 (E.L.S.)], and Harvard University John A. Paulson School of Engineering and Applied Sciences (C.J.W.).

## Author contributions

Conceptualization: Y.J. and C.J.W. Methodology: Y.J., J.T.A., E.L.S., and C.J.W. Investigation: Y.J., J.T.A., K.S., A.C., and C.J.W. Analysis: Y.J., J.T.A., K.S., A.C., and C.J.W. Visualization: Y.J., J.T.A., and U.S.C. Writing – original draft: Y.J. and C.J.W. Writing – review and editing: Y.J., J.T.A., E.L.S., K.S., U.S.C., R.W.N., R.D.H., and C.J.W. Supervision: R.W.N., R.D.H., and C.J.W. Funding acquisition: R.D.H. and C.J.W.

## Competing interests

The authors declare no competing interests.
