## [Peer Review File · Nature Communications]

REVIEWER COMMENTS

Reviewer #1 (Remarks to the Author):

The proposed approach is very interesting, however I noted two main problems for this study:

1) Only 8 subjects have been involved, and in some experiments (dumbbell curl and treadmill/outdoor locomotion test) just one participant was tested

2) the (inter- and intra-operator) reliability of measures with respect to Sensor placements were not assessed.

Another minor comment regards the Inverse kinematics, that is often used in biomechanical studies, so this approach deserves to be better explained in Introduction

Reviewer #2 (Remarks to the Author):

The Authors illustrate a method for estimating the equivalent torque generated at the level of simple joints in a minimally invasive way, i.e. without resorting to laboratory set-ups.

The basic idea is to find a correlation between the increase in the thickness of the involved primary muscle and the torque delivered at joint level. To this end, they use amplitude mode (A-mode) ultrasonic transducers, which are known to be better suited to the purpose as they are simpler and more portable than brightness mode (B-mode) transducers.

It is well known that 1D A-mode ultrasound scans can be used to measure the thickness of soft tissues, e.g. a muscle. Therefore the objective is to show how it is possible to correlate the variation in thickness of a muscle with the torque applied to a joint.

The experiments are conducted on two joints with superficial primary muscles: the elbow (biceps brachii, BB) and the knee (rectus femoris, RF).

The Authors correctly point out that the evaluation of joint torques can provide useful information for estimating the risk of injury in sport or for evaluating the progress of a rehabilitation therapy. This is

certainly true. They can also add that this evaluation would have an important impact in the field of ergonomics for the prevention of work-related injuries, which have an even greater socio-economic impact.

The authors rightly mention the limitations of EMG saying that "EMG measures the electrical activation of a muscle, representing neurological input rather than the muscle's mechanical output". However, they do not mention the Mechanomyogram (MMG), which instead provides a signal in response to actual muscle contraction.

Resistive elastic bands, which can be used to determine the change in shape of a superficial muscle, such as BB or RF, can also provide information that follows neurological input. Given that they are limiting themselves to superficial primary muscles, it cannot be excluded that conductive elastic bands can also provide an equally informative signal, especially in conjunction with IMUs and using data fusion techniques, which do not necessarily require computationally expensive methods.

The Authors want to demonstrate the effectiveness of their algorithm (MBTA) to determine muscle thickness in a simple way. But the analysis is not conducted in real-time, but rather in post-processing. Therefore, the doubt remains as to the possibility of using this approach to monitor the torque delivered during the execution of daily tasks. Surely the need for manual image segmentation does not help.

It is unclear how the MBTA algorithm might work in the case of deep muscles.

The use of different ultrasonic elements is not new and the performance seems not to improve previous results, e.g. Tsutsui, Y., Tanaka, T., Kaneko, S.I. and Feng, M.Q., 2005, October. Duplex ultrasonic muscle activity sensor. In *SENSORS, 2005 IEEE* (pp. 4-pp). IEEE.

Validation of the algorithm requires further experimental work. In fact, the tests on elbow flexion and on walking were conducted on a single subject. How can this result be considered significant? And why a single subject was involved in some experiments, while 8 in others?

How does the algorithm work in the case of deeper muscles or in the case of ball-socket joints?

The article's approach is rather empirical. The basic idea is to look for a correlation between the variation in thickness of a muscle and the equivalent torque applied to a joint. For example, $(MT+Ang)^2$ was used as one of the input variable for fitting. Using the sum of heterogeneous quantities is not a robust choice, because the result would depend from the used units. Also, a quadratic relationship is used "because it could capture the commonly observed curvilinear relationship between muscle deformation and contraction intensity". This is a rather trenchant approach. Is it a second order approximation of any non-linear response? What about the use of exponential functions or higher-order polynomials? In my

opinion, the most elegant approach would have been starting from biomechanical muscle contraction models.

MINOR DETAILS

- The expression "radio frequency" applied to ultrasound waves is a bit of a misnomer, which unfortunately is used by some authors.
- The 4 transducers are inclined at a certain angle, and the reader is referred to Fig. 1. But in fig. 1 it is not clear that they are. The lines representing the SET axes should probably be slanted.
- 4 SETs are used to find the one with the most intense echo. Would 3 suffice? Would 5 or 6 be better? In short, how was the number of SETs determined?
- The model of the rechargeable battery pack ("RRC, Germany") should be inserted in Methods.
- In "Experimental methods and data processing" the arm dominance should be declared (now it is stated: "regardless of the dominance")
- In "Dynamometer testing" it is stated that gravitational torques are modelled as "a linear function to joint angle". This not clear. I would expect a trigonometric function, similar to that in Eq. 1.
- In "Dumbbell curl rigid body model" L is the distance from the elbow to the hand. Since the hand has its own length, this is a generic definition. Authors should better indicate the anatomical reference points used to assess the lengths.
- Equation 1 assumes that the centre of gravity lies in the middle of L. Why?

Reviewer #2 (Remarks on code availability):

The code is complete and clear. Data are available and results reproducible.

Reviewer #3 (Remarks to the Author):

The manuscript "Estimation of Joint Torque in Dynamic Activities Using Wearable A-Mode Ultrasound" reports a dynamic muscle thickness extraction and joint torque estimation method based on wearable A-mode ultrasound. The study proposes a multi-view A-mode ultrasound sensing solution and a hybrid muscle thickness tracking method for robust extraction of muscle thickness in dynamic motions, as well as a second-order polynomial fitting method that integrates joint kinematic information for better joint

torque estimation. Exhaustive simulations and experiments were performed to validate the effectiveness of the proposed method, with very promising results achieved. This work will contribute to an in-depth understanding of muscle activity during dynamic and unconstrained movements and holds great promise for biomechanics and rehabilitation robotics research.

1. The main innovation of this paper is to achieve accurate muscle thickness tracking using wearable A-mode ultrasound through a multi-view sensing strategy and a hybrid muscle thickness extraction algorithm. While the algorithm works well for offline analysis, I am interested in how the algorithm can be used for real-time applications (e.g., real-time estimation of joint torque for robot control). As a lightweight sensing strategy, it will be more interesting to combine A-mode ultrasound with wearable robotics, which requires real-time computational power of the algorithms. Offline analysis is also important for certain biomechanical studies, however, B-mode ultrasound can also work well in this case.

2. Regarding the muscle boundary tracking algorithm, the authors state that the sensor with the strongest echoes is used when measuring muscle thickness. This sensor selection method helps to select the sensor with the least rotation to the target muscle. However, during the muscle contraction, the deformation of the muscle changes the relative angle between the muscle and the sensor, which means that the optimal sensing angle is constantly changing during muscle contraction. How did the authors solve this problem? For offline analysis, some filtering methods can help smooth out the results, even if there is some jitter. However, this can cause problems for online applications. In terms of the simulation study in Fig. S3, what is the amplitude of the sine wave?

3. For the comparison of A-mode and B-mode ultrasound for muscle thickness tracking in Fig. S4, the processing of B-mode ultrasound is still in an A-mode way. The authors selected some columns from the B-mode ultrasound image to simulate the use of A-mode ultrasound sensing. Since it is comparative analysis here, why not extract the muscle thickness from the B-mode to provide a better ground truth?

4. There are quite a few single case experiments in the paper, such as dumbbell curls and treadmill/outdoor locomotion tests. These experiments make more sense to the reader, and it would be nice to extend them a bit. In particular, for the dumbbell curl experiment (Fig. 3), I am supervised that the elbow torque estimated by the A-mode ultrasound matched the simple simulation model quite well.

5. In terms of second-order polynomial fitting, it is basically similar to the linear regression applying a second-order polynomial basis function. Incorporating joint kinematics into the model also makes sense because both muscle activation (thickness in this case) and muscle length (joint kinematics) affect the joint torque. When building the polynomial fitting model, did the authors use the data covering all the different contraction tasks? If so, the model might be overfitted because all the data came from

repeatable movements. It is better to test the generalisability of the model by training and testing it on different contraction tasks.

6. In the locomotion test, the authors took the hip-knee angle difference and the thigh angle input kinematic variables. What is the difference between thigh angle and hip angle? I recommend inputting hip and knee angles into the model because the model will handle the difference if necessary.

7. In Fig.1 D, the red line doesn't appear to match the strongest echo in the signal.

Reviewer #1

The proposed approach is very interesting, however I noted two main problems for this study:

1) Only 8 subjects have been involved, and in some experiments (dumbbell curl and treadmill/outdoor locomotion test) just one participant was tested.

We thank the reviewer for their constructive feedback regarding the need for more participants to improve the robustness of the work. In response, we have increased the number of participants from 8 to 10 for the dynamometer tests and from 1 to 5 for the dumbbell curl and treadmill locomotion tests. With the newly added participants, we now have similar participant pools to other relevant papers in the field. For example:

- N = 5 in Martin, Jack A., et al. "Gauging force by tapping tendons." *Nature communications* 9.1 (2018): 1592.
- N = 8 in Alvarez, Jonathan T., et al. "Towards soft wearable strain sensors for muscle activity monitoring." *IEEE Transactions on Neural Systems and Rehabilitation Engineering* 30 (2022): 2198-2206.
- N = 6 in Zhou, Yu, et al. "Bio-signal based elbow angle and torque simultaneous prediction during isokinetic contraction." *Science China Technological Sciences* 62 (2019): 21-30.

We have maintained N=1 for the outdoor locomotion test. We justify this choice based on the absence of ground truth in outdoor environments and the primary objective of this test, which is to demonstrate applicability in real-world settings rather than quantitative validation of the proposed method (which is the goal for the treadmill part with N=5).

With the added participants, we obtained results comparable to the ones from the original submission. We refer the reviewer to the revised manuscript with highlighted changes for a comprehensive overview but have included a few call outs below.

[Updated Text]

Results

Elbow torque estimation during isokinetic contractions

To evaluate the estimation of corresponding joint torque from parallel muscles, we acquired ... on **ten** healthy adults while performing ...

...

Across all participants, the individualized models achieved RMSEs of **3.89 ± 0.86** Nm (mean \pm SD), NRMSEs of **$7.6 \pm 1.4\%$** , and coefficients of determination (R^2) of 0.92 ± 0.03 (table S1).

Elbow torque estimation during dumbbell curls

To investigate functional applications of elbow torque estimation, **we performed a study on five participants during dumbbell curls.**

...

For all participants, there was close agreement between the estimated and calculated torque (Fig. S8), with the average elbow torque increasing proportionally to the weight of the dumbbell (Fig. 3e, Fig. 3f). Quantitatively, absolute errors in average torque measurements were 1.25 ± 0.71 Nm for no weight, 1.18 ± 0.57 Nm for medium weights, and 2.41 ± 0.94 Nm for heavy weights across all participants. Qualitatively, ...

Knee torque estimation during isokinetic contractions

To evaluate the estimation of corresponding joint torque from pennate muscles, we collected ... on ten participants during isokinetic RF contractions.

...

By fitting $(MT, \text{Ang})^2$ on data from each participant across all contractions (Fig. 4c), we obtained individualized models with RMSEs of 12.59 ± 3.12 Nm, NRMSEs of $7.0 \pm 1.3\%$, and R^2 of 0.92 ± 0.02 (table S1).

Knee torque estimation during treadmill and outdoor locomotion

To evaluate knee torque estimation during unconstrained dynamic activities, we performed a treadmill study on five participants and a single-participant outdoor locomotion demonstration.

...

By fitting quadratic fits on data from each participant across all treadmill conditions, we obtained individualized models with RMSEs of 17.43 ± 5.09 Nm, NRMSEs of $6.0 \pm 1.1\%$, and R^2 of 0.92 ± 0.03 (table S2).

[Updated Figures]

Results

Elbow torque estimation during dumbbell curls

(We included the following subfigures in Fig. 3 to show the results from all 5 participants.)

Fig. 3. Elbow torque estimation during dumbbell curls. (E) Average estimated torque for all participants ($n = 5$) at different weight conditions. Each color represents data for one participant. Each dot is obtained by taking the average of all contractions ($n = 6$) within the respective condition. (F) Average calculated torque for all participants ($n = 5$) at different weight conditions.

Knee torque estimation during treadmill and outdoor locomotion

(We updated Fig. 5c and Fig. 5d to show the effects of locomotion type and slope level on RMSEs for all 5 participants. Also, we changed the plots from bar plots to box plots to better show the underlying data distribution.)

Fig. 5. Knee torque estimation during treadmill and outdoor locomotion. (C) Distributions of RMSEs for different locomotion types across all participants (n = 5). Colored dots represent the RMSE for each participant. (D) Distributions of RMSEs for different slope levels across all participants (n = 10).

Supplementary Information

(We added the following supplementary figure to show data for all participants during the dumbbell curl test.)

Fig. S8: Data from different participants during the dumbbell curl study. Each row represents BB thickness (left column), elbow angle (center left column), estimated elbow torque (center right column), and calculated elbow torque from a rigid body model (right column) for one participant during curls with different dumbbell weights. Data from Participant 1 is shown in Fig. 3. Lines and shaded regions represent mean \pm SD (n = 6 repetitions).

2) the (inter- and intra-operator) reliability of measures with respect to Sensor placements were not assessed.

We thank the reviewer for pointing out the need for such a reliability assessment. We fully agree that it is important to evaluate the effect of sensor placement to better understand the method's performance during realistic applications. In response, we conducted a single-participant sensitivity analysis on transducer placement and have added the following text and figure to show the analysis results.

[Updated Text]

Results

Knee torque estimation during isokinetic contractions

Lastly, we performed a single-participant sensitivity analysis to understand the effect of transducer placement (Supplementary Text). By collecting A-mode ultrasound at 18 different locations across the RF, we observed that transducer placement largely affected the ultrasound quality of the deep RF boundary but did not dramatically affect the correlation to knee torque (Fig. S9). However, applying the fit obtained from a specific location to neighboring locations led to increasing estimation errors, highlighting the importance of minimizing sensor placement drift during use.

[Updated Figure]

Supplementary Information

Fig. S9: Sensitivity analysis on transducer placement. (A) SETs were placed on 18 different locations (3 by 6 grid) across the RF muscle. At each location, the participant performed 4 repetitions of passive, concentric, and eccentric knee extensions on the dynamometer. (B) Muscle boundary prominence scores from all tested locations. This score is designed to quantitatively assess the ultrasound signal quality of the deep RF muscle boundary. Specifically, this score is calculated by averaging the maximum peak prominence of the raw ultrasound data within the region of interest across all times frames and then normalizing the scores from all locations. (C) Coefficients of determination scores for (MT, Ang)² models from all locations. Each model was obtained using the ultrasound and dynamometer data collected at the respective location. (D) Normalized RMSEs obtained by fitting a

(MT, Ang)² model on the reference location (starred in **B**, **C**, and **D**) and evaluated on the 8 neighboring locations.

Another minor comment regards the Inverse kinematics, that is often used in biomechanical studies, so this approach deserves to be better explained in Introduction

Thank you for your suggestion. We have indeed used inverse dynamics to obtain ground truth knee torque in our treadmill locomotion test and agreed that it should be better introduced. We have updated the Introduction, adding a better description of the technique.

[Old Text]

Introduction

Modeling techniques (e.g., inverse dynamics, computational musculoskeletal models)^{16,17} and surface electromyography (EMG)^{18,19} have traditionally been used to indirectly estimate both muscle force and joint torque. However, simulation models rely on a combination of kinematic and lab-based kinetic measurements, limiting their use to primarily research environments⁹.

[Updated Text]

Introduction

Modeling techniques (e.g., inverse dynamics, computational musculoskeletal models)^{16,17} have commonly been used to indirectly estimate both muscle force and joint torque. **Specifically, inverse dynamics can reliably estimate joint torques within a rigid body model using motion and external force measurements. With additional optimization and biomechanical constraints, musculoskeletal models can further estimate individual muscle forces. Despite these simulation models' wide use in biomechanical studies, they rely on intensive computation with a combination of measurements (e.g., motion-capture systems, force platforms) that are yet largely constrained to research environments⁹.**

Reviewer #2

The Authors illustrate a method for estimating the equivalent torque generated at the level of simple joints in a minimally invasive way, i.e. without resorting to laboratory set-ups.

The basic idea is to find a correlation between the increase in the thickness of the involved primary muscle and the torque delivered at joint level. To this end, they use amplitude mode (A-mode) ultrasonic transducers, which are known to be better suited to the purpose as they are simpler and more portable than brightness mode (B-mode) transducers.

It is well known that 1D A-mode ultrasound scans can be used to measure the thickness of soft tissues, e.g. a muscle. Therefore, the objective is to show how it is possible to correlate the variation in thickness of a muscle with the torque applied to a joint.

The experiments are conducted on two joints with superficial primary muscles: the elbow (biceps brachii, BB) and the knee (rectus femoris, RF).

Thank you for your thoughtful and thorough overview of this work. We appreciate the comments and suggestions you provided!

The Authors correctly point out that the evaluation of joint torques can provide useful information for estimating the risk of injury in sport or for evaluating the progress of rehabilitation therapy. This is certainly true. They can also add that this evaluation would have an important impact in the field of ergonomics for the prevention of work-related injuries, which have an even greater socio-economic impact.

Thank you for your insightful comment and pointing out the potential implication of joint torque estimation in ergonomics. As rightly pointed out by the reviewer, there are substantial socio-economic impacts associated with work-related injuries. To address your suggestion, we have edited the Discussion of the manuscript by including the following sentence and references.

[Updated Text]

Discussion

Moreover, given the prevalence of work-related lateral epicondylitis (tennis elbow)⁶⁰ and occupational knee disorders⁶¹, tracking elbow and knee torque while conducting physically demanding jobs could also help facilitating better workplace ergonomics and preventing work-related injuries.

References

60. Bretschneider, S. F., Los, F. S., Eygendaal, D., Kuijer, P. P. F. M. & Molen, H. F. van der. Work-relatedness of lateral epicondylitis: Systematic review including meta-analysis and GRADE work-relatedness of lateral epicondylitis. *Am. J. Ind. Med.* 65, 41–50 (2022).
61. Reid, C. R., Bush, P. M., Cummings, N. H., McMullin, D. L. & Durrani, S. K. A Review of Occupational Knee Disorders. *J. Occup. Rehabilitation* 20, 489–501 (2010).

The authors rightly mention the limitations of EMG saying that "EMG measures the electrical activation of a muscle, representing neurological input rather than the muscle's mechanical output". However, they do not mention the Mechanomyogram (MMG), which instead provides a signal in response to actual muscle contraction.

We thank the reviewer for their comment and appreciate the opportunity to enrich the background information presented in the Introduction. In response to your comment, we have included the following sentences and references to describe MMG in the context of this work.

[Updated Text]

Introduction

Mechanomyography (MMG) estimates muscle mechanical loads by measuring the lateral oscillations elicited from contracting muscle fibers. Despite being considered as the mechanical counterpart to EMG, MMG is highly susceptible to motion artifact from limb movement, especially during dynamic activities^{23,24}.

References

23. Ibitoye, M. O., Hamzaid, N. A., Zuniga, J. M. & Wahab, A. K. A. Mechanomyography and muscle function assessment: A review of current state and prospects. *Clin Biomech* 29, 691–704 (2014).

24. Posatskiy, A. O. & Chau, T. The effects of motion artifact on mechanomyography: A comparative study of microphones and accelerometers. *J. Electromyogr. Kinesiol.* 22, 320–324 (2012).

Resistive elastic bands, which can be used to determine the change in shape of a superficial muscle, such as BB or RF, can also provide information that follows neurological input. Given that they are limiting themselves to superficial primary muscles, it cannot be excluded that conductive elastic bands can also provide an equally informative signal, especially in conjunction with IMUs and using data fusion techniques, which do not necessarily require computationally expensive methods.

We thank the reviewer for their insightful comment for a more comprehensive review of existing technologies. We agree with the reviewer that there exist many exciting works aiming to capture muscle shape changes with affordable and computationally lightweight solutions. Sensorized elastic bands or force myography (FMG) being one of the elegant solutions. However, such sensing bands are often wrapped around the entire limb of interest, capturing signals from all agonist-antagonist muscle pairs within the limb (e.g., shape changes from both the elbow flexors and extensors). Sensitive soft strain sensors have been recently proposed to capture the localized skin deformation, but still lack the specificity between superficial and deep muscles (e.g., combined shape changes from the biceps brachii and the underneath brachialis). We believe that ultrasound holds a unique advantage of providing muscle specific measurements, which enabled our work of evaluating the correlation between A-mode captured RF and BB deformations to knee and elbow torques. To capture this information, we have modified the Introduction with new sentences describing FMG. Once again, thank you for the opportunity to articulate this more clearly.

[Old Text]

Introduction

Ultra-sensitive soft strain sensors adhered to skin can non-invasively capture the underlying muscle bulge, which has been found to positively correlate with changes in joint torque during dynamometer tests²⁴. However, this surface-level measurement cannot distinguish between deformations from superficial and deep muscles.

[Updated Text]

Introduction

Force myography (FMG) captures the muscle geometry change by measuring pressure variations on the skin that occurs from muscle bulging. By using a band of pressure sensors wrapped around the limb, FMG captures the net shape change from all muscles within the limb, often including agonist and antagonist muscle pairs²⁵. More recently, ultra-sensitive soft strain sensors have been adhered to skin to capture the localized skin deformation from the underlying muscle bulge, which has been found to positively correlate with changes in joint torque during dynamometer tests²⁶. However, this surface-level measurement cannot **decouple** deformations from superficial and deep muscles.

References

25. Xiao, Z. G. & Menon, C. A Review of Force Myography Research and Development. *Sensors Basel Switz* 19, 4557 (2019).

The Authors want to demonstrate the effectiveness of their algorithm (MBTA) to determine muscle thickness in a simple way. But the analysis is not conducted in real-time, but rather in post-processing. Therefore, the doubt remains as to the possibility of using this approach to monitor the torque delivered during the execution of daily tasks. Surely the need for manual image segmentation does not help.

Thank you for your interest in the real-time aspect of our algorithm. Indeed, both the MBTA and torque estimation are currently conducted in post-processing. However, we believe in their potential for real-time implementation, given the simplicity of the involved processing. Specifically, it would be interesting for future research to replace the manual image segmentation with an automatic calibration process that determines the region of interesting based on an individual's weight and height. In response, we have updated our writing to highlight the method's potential for real-time application.

[Updated Text]

Discussion

Third, while the MBTA demonstrates potential for real-time implementation (Methods), current torque estimation is conducted in post-processing. Future efforts could focus on developing a centralized system to simultaneously log ultrasound and IMU data and perform the torque estimation algorithm in real-time.

Methods

Muscle boundary tracking algorithm (MBTA)

Although the MBTA is currently implemented in post-processing, its required computational steps are straightforward. The algorithm operates with an average runtime of less than 0.3 milliseconds per frame on a standard CPU (Intel Core i7-10750H @ 2.60 GHz), showcasing its potential for real-time use. To transition to real-time implementation, further developments could involve pre-defining the region of interest based on body measures, such as weight and height, replacing low-pass filters with the simpler moving average filters, and implementing the algorithm on a field programmable gate array (FPGA). Notably, the 2D box filter in the brightness-based method and the smoothing filters in the final fusion step may introduce time delays in tracking. Therefore, it is important to optimize filter parameters, like filter size and cutoff frequency, for specific applications to minimize these delays.

It is unclear how the MBTA algorithm might work in the case of deep muscles.

Thank you for your interest in the MBTA algorithm. Indeed, the MBTA can track the boundaries of deep muscles, such as the vastus intermedius (one of the deeper thigh muscles). We have revised our writing and included a subfigure in Fig. S1 to show the MBTA's applicability for deep muscles.

[Updated Text]

Results

Muscle thickness tracking during motion

The MBTA can continuously track thickness changes in both superficial and deep muscles during contractions (Fig. S1).

[Updated Figure]

Supplementary Information

Fig. S1: Overview of the muscle boundary tracking algorithm (MBTA). (F) MBTA-traced boundaries of the biceps brachii (BB), rectus femoris (RF), and vastus intermedius (VI) (red lines) overlaid on A-mode ultrasound data from (A)

The use of different ultrasonic elements is not new and the performance seems not to improve previous results, e.g. Tsutsui, Y., Tanaka, T., Kaneko, S.I. and Feng, M.Q., 2005, October. Duplex ultrasonic muscle activity sensor. In SENSORS, 2005 IEEE (pp. 4-pp). IEEE.

We thank the reviewer for their comment and the opportunity to further clarify the objectives and contributions of our work. We agree with the reviewer that there has been prior work on using multiple single-element transducers to evaluate muscle functions. However, our study aims not to present a new way for fusing signals from multiple transducers, but rather to investigate the applicability of A-mode ultrasound for joint torque estimation during various dynamic activities (which required us to use multiple transducers to ensure signal quality). We have modified the Introduction to articulate this objective more clearly.

We have also thoroughly read the publication you listed. While they had very nice results, they were generated from a single participant during static isometric knee extensions. We evaluated multiple participants (N=10) during isokinetic elbow flexions and knee extensions at various speeds. In addition, we also evaluated the effectiveness of our method with 5 participants during unconstrained dynamic activities of dumbbell curls and treadmill and outdoor locomotion. However, we appreciate the exploration by Tsutsui et al., as it is one of the earliest works in wearable A-mode ultrasound and have included their work as an additional reference in our paper.

[Old Text]

Introduction

Only a few studies have investigated estimating hand grip force during static grips^{43,44} or elbow torque during controlled isokinetic elbow flexions⁴⁵. The use of A-mode ultrasound for reliable muscle

deformation measurement and joint torque estimation during dynamic and unconstrained functional tasks remains unexplored.

[Updated Text]

Introduction

Only a few studies have investigated estimating hand grip force^{45,46} and **knee torque⁴⁷ during static isometric contractions** or elbow torque during controlled isokinetic elbow flexions⁴⁸. The use of A-mode ultrasound for reliable muscle deformation measurement and joint torque estimation during dynamic and unconstrained functional tasks remains unexplored. **Hence, it remains a gap to investigate the applicability of A-mode ultrasound for continuous tracking of muscle mechanical loads during unconstrained dynamic activities in real-world environments.**

References

47. Tsutsui, Y., Tanaka, T., Kaneko, S. & Feng, M. Q. Duplex Ultrasonic Muscle Activity Sensor. *IEEE Sens.*, 2005 310–313 (2005) doi:10.1109/icsens.2005.1597698.

Validation of the algorithm requires further experimental work. In fact, the tests on elbow flexion and on walking were conducted on a single subject. How can this result be considered significant? And why a single subject was involved in some experiments, while 8 in others?

We thank the reviewer for pointing out the need for more participants. We originally designed the dynamometer tests (N = 8) as the main evaluation for this work and the remaining single-participant tests (dumbbell curl and treadmill/outdoor locomotion) as demonstrations. However, we acknowledge the lack of participants for making claims regarding the method's effectiveness in unconstrained dynamic activities. In response, we have now extended the participant pool from 8 to 10 for the dynamometer tests and from 1 to 5 for the dumbbell curl and treadmill locomotion tests. With the newly added participants, we now have similar participant pools to other relevant papers in the field. For example:

- N = 5 in Martin, Jack A., et al. "Gauging force by tapping tendons." *Nature communications* 9.1 (2018): 1592.
- N = 8 in Alvarez, Jonathan T., et al. "Towards soft wearable strain sensors for muscle activity monitoring." *IEEE Transactions on Neural Systems and Rehabilitation Engineering* 30 (2022): 2198-2206.
- N = 6 in Zhou, Yu, et al. "Bio-signal based elbow angle and torque simultaneous prediction during isokinetic contraction." *Science China Technological Sciences* 62 (2019): 21-30.

We have maintained N=1 for the outdoor locomotion test. We justify this choice based on the absence of ground truth in outdoor environments and the primary objective of this test, which is to demonstrate applicability in real-world settings rather than quantitative validation of the proposed method (which is the goal for the treadmill part with N=5).

We refer the reviewer to the revised manuscript with highlighted changes for a comprehensive overview of all changes but have included a few call outs below.

[Updated Text]

Results

Elbow torque estimation during isokinetic contractions

To evaluate the estimation of corresponding joint torque from parallel muscles, we acquired ... on **ten** healthy adults while performing ...

...

Across all participants, the individualized models achieved RMSEs of 3.89 ± 0.86 Nm (mean \pm SD), NRMSEs of $7.6 \pm 1.4\%$, and coefficients of determination (R^2) of 0.92 ± 0.03 (table S1).

Elbow torque estimation during dumbbell curls

To investigate functional applications of elbow torque estimation, we performed a study on five participants during dumbbell curls.

...

For all participants, there was close agreement between the estimated and calculated torque (Fig. S8), with the average elbow torque increasing proportionally to the weight of the dumbbell (Fig. 3e, Fig. 3f). Quantitatively, absolute errors in average torque measurements were 1.25 ± 0.71 Nm for no weight, 1.18 ± 0.57 Nm for medium weights, and 2.41 ± 0.94 Nm for heavy weights across all participants. Qualitatively, ...

Knee torque estimation during isokinetic contractions

To evaluate the estimation of corresponding joint torque from pennate muscles, we collected ... on **ten** participants during isokinetic RF contractions.

...

By fitting $(MT, \text{Ang})^2$ on data from each participant across all contractions (Fig. 4c), we obtained individualized models with RMSEs of 12.59 ± 3.12 Nm, NRMSEs of $7.0 \pm 1.3\%$, and R^2 of 0.92 ± 0.02 (table S1).

Knee torque estimation during treadmill and outdoor locomotion

To evaluate knee torque estimation during unconstrained dynamic activities, we performed a treadmill study on five participants and a single-participant outdoor locomotion demonstration.

...

By fitting quadratic fits on data from each participant across all treadmill conditions, we obtained individualized models with RMSEs of 17.43 ± 5.09 Nm, NRMSEs of $6.0 \pm 1.1\%$, and R^2 of 0.92 ± 0.03 (table S2).

[Updated Figures]

Results

Elbow torque estimation during dumbbell curls

(We included the following subfigures in Fig. 3 to show the results from all 5 participants.)

Fig. 3. Elbow torque estimation during dumbbell curls. (E) Average estimated torque for all participants ($n = 5$) at different weight conditions. Each color represents data for one participant. Each

dot is obtained by taking the average of all contractions ($n = 6$) within the respective condition. (F) Average calculated torque for all participants ($n = 5$) at different weight conditions.

Knee torque estimation during treadmill and outdoor locomotion

(We updated Fig. 5c and Fig. 5d to show the effects of locomotion type and slope level on RMSEs for all 5 participants. Also, we changed the plots from bar plots to box plots to better show the underlying data distribution.)

Fig. 5. Knee torque estimation during treadmill and outdoor locomotion. (C) Distributions of RMSEs for different locomotion types across all participants ($n = 5$). Colored dots represent the RMSE for each participant. (D) Distributions of RMSEs for different slope levels across all participants ($n = 10$).

Supplementary Information

(We added the following supplementary figure to show data for all participants during the dumbbell curl test.)

Fig. S8: Data from different participants during the dumbbell curl study. Each row represents BB thickness (left column), elbow angle (center left column), estimated elbow torque (center right column), and calculated elbow torque from a rigid body model (right column) for one participant during curls with different dumbbell weights. Data from Participant 1 is shown in Fig. 3. Lines and shaded regions represent mean \pm SD (n = 6 repetitions).

How does the algorithm work in the case of deeper muscles or in the case of ball-socket joints?

Thank you for your interest in extending our work beyond the presented muscles and joints. As responded to your 5th comment, the MBTA can track the thickness of deep muscles (e.g., the VI). While we are intrigued by the potential of A-mode-based torque estimation in deeper muscles and more complex joints, we believe that these aspects are beyond the scope of the current work. In response, we have modified the Discussion by clarifying the scope of this work and outlining promising future directions regarding more diverse muscles and joints.

[Updated Text]

Discussion

Lastly, our study focused primarily on superficial muscles (e.g., BB and RF) and hinge joints (e.g., elbow and knee). While the MBTA can track the thickness of deep muscles (Fig. S1), using deep muscle dynamics for estimating joint torque, especially in more complex ball-and-socket joints, remains interesting and unexplored.

The article's approach is rather empirical. The basic idea is to look for a correlation between the variation in thickness of a muscle and the equivalent torque applied to a joint. For example, $(MT+Ang)^2$ was used as one of the input variable for fitting. Using the sum of heterogeneous quantities is not a robust choice, because the result would depend from the used units. Also, a quadratic relationship is used "because it could capture the commonly observed curvilinear relationship between muscle deformation and contraction intensity". This is a rather trenchant approach. Is it a second order approximation of any non-linear response? What about the use of exponential functions or higher-order polynomials? In my opinion, the most elegant approach would have been starting from biomechanical muscle contraction models.

We thank the reviewer for their comment, which helped us to improve the clarity of the paper. To clarify, we intended to apply the quadratic fit on muscle thickness (MT) and joint kinematics (Ang), as two independent input variables, rather than on the sum of the two variables. We recognize that the notation, $(MT+Ang)^2$, might have been confusing. We have changed it to $(MT, Ang)^2$ and have updated the relevant text to avoid potential confusion.

Additionally, we agree with the reviewer that our approach in identifying the model function has been empirical. We also fully agree that deriving the function from a commonly accepted biomechanical model would have been a more elegant and better approach. However, to our knowledge, it has been challenging to develop accurate and accessible biomechanical models on muscle 3D deformation. Therefore, existing work has explored a spectrum of nonlinear functions, including quadratic, cubic, exponential, and even more advanced machine learning models. In our work, we chose the quadratic

fit mainly for its simplicity. However, we acknowledge the empirical nature of this approach and have included a new subsection in the Methods to clarify these design decisions.

[Old Text]

Results

Elbow torque estimation during isokinetic contractions

For each participant, we applied a quadratic fit to data from all conditions with BB thickness and elbow angle as input variables and elbow torque as the target variable $[(MT+Ang)^2]$ (Fig. 2c). We chose the 2nd order quadratic fit because it could capture the commonly observed curvilinear relationship between muscle deformation and contraction intensity^{26,27,46,48,51}.

[Updated Text]

Results

Elbow torque estimation during isokinetic contractions

For each participant, we applied a quadratic fit to data from all conditions (Methods). The quadratic fit, denoted as $(MT, Ang)^2$, uses two input variables, BB thickness and elbow angle, and estimates elbow torque as the target variable (Fig. 2c).

Methods

Quadratic fit on muscle thickness and joint kinematics

The non-linear relationship between muscle deformation and force production has been widely reported in literature^{28,49,51}. However, it remains challenging to develop accurate and accessible biomechanical models that comprehensively capture the muscle's complex 3D deformation during contraction⁷¹, not to mention the interactions with neighboring muscles⁶⁴. In practice, this challenge has led to a rather empirical approach in defining the relationship between muscle deformation measurements and joint torque estimates. Specifically, a spectrum of models, including quadratic²⁹, cubic²⁶, exponential⁷², and even more complex machine learning⁴⁵⁻⁴⁸ models have been suggested to describe such relationship. In this work, we chose the quadratic fit, mainly for its simplicity, to describe the mapping from muscle thickness and joint kinematics to joint torque.

MINOR DETAILS

- The expression "radio frequency" applied to ultrasound waves is a bit of a misnomer, which unfortunately is used by some authors.

We thank the reviewer for this important correction. We have revised our terminology and removed the term "radio frequency" in reference to ultrasound signals throughout the manuscript.

- The 4 transducers are inclined at a certain angle, and the reader is referred to Fig. 1. But in fig. 1 it is not clear that they are. The lines representing the SET axes should probably be slanted.

Thank you for your detailed observation. We have updated Fig. 1a by slanting the transducers to have a more realistic representation of the system.

Old Fig. 1a

Updated Fig. 1a

- 4 SETs are used to find the one with the most intense echo. Would 3 suffice? Would 5 or 6 be better? In short, how was the number of SETs determined?

Thank you for your question regarding the number of SETs used in our study. In response, we have added the below sentence in the Methods to clarify our design decision. In summary, using more transducers at a larger range of angles increases the likelihood of capturing data with strong echoes. However, it also leads to a larger transducer assembly and could potentially decrease the frame rate (assuming each SET operates sequentially rather than in parallel). In our work, we empirically determined that four SETs were sufficient based on our preliminary testing.

[Updated Text]

Methods

Amplitude-mode ultrasound system

The choice of four transducers was based on preliminary tests as this number provided a good balance between capturing signals with sufficient echo strength and achieving reasonable technical specifications including assembly size and frame rate (90 Hz).

- The model of the rechargeable battery pack (“RRC, Germany”) should be inserted in Methods.

Thank you for catching this. We have modified the relevant section in Methods to include the model of the battery.

[Updated Text]

Methods

Amplitude-mode ultrasound system

For system wearability, we powered the ultrasound instrument with a rechargeable battery pack (RRC2054, RRC Power Solutions, Germany) and housed all electronics in a backpack (Fig. 5e).

- In "Experimental methods and data processing" the arm dominance should be declared (now it is stated: "regardless of the dominance")

Thank you for pointing this out. To avoid confusion, we have removed the phrase "regardless of the dominance". Additionally, we have included two new tables, Table S3 and Table S4, to document the participant information for the different experiments. These tables also declare the dominant sides of all participants.

[Updated Text]

Supplementary Information

Table S3: Participant information for dynamometer tests.

	Gender	Age	Height (cm)	Mass (kg)	Dominant side
Participant 1	F	26	162	58	Right
Participant 2	F	29	160	59	Right
Participant 3	M	29	183	83	Right
Participant 4	M	33	177	77	Right
Participant 5	F	28	164	52	Right
Participant 6	M	27	180	88	Left
Participant 7	M	37	184	77	Right
Participant 8	M	25	189	80	Right
Participant 9	M	30	181	70	Right
Participant 10	M	26	170	62	Right

Table S4: Participant information for dumbbell curl and treadmill locomotion tests.

	Gender	Age	Height (cm)	Mass (kg)	Dominant side
Participant 1	M	29	183	83	Right
Participant 2	M	24	193	84	Left
Participant 3	M	27	180	88	Left
Participant 4	M	30	181	70	Right
Participant 5	F	29	160	59	Right

- In "Dynamometer testing" it is stated that gravitational torques are modelled as "a linear function to joint angle". This not clear. I would expect a trigonometric function, similar to that in Eq. 1.

Thank you for catching this. We have updated the gravitational torque correction model from a linear function to a sinusoidal function and re-run all analyses accordingly. Although this update did not alter the previously reported results much, we agree that the sinusoidal function is a more accurate representation of the gravitational torque in dynamometer readings.

[Old Text]

Methods

Experimental methods and data processing

To isolate biological torque changes, we first modeled gravitational torque (torque measurements in passive contractions) as a linear function to joint angle and subsequently subtracted this function from torque measurements in all passive and active contractions.

Supplementary Information

Table S1: Estimation accuracies for (MT+Ang)² models in isokinetic dynamometer tests.

Metric	Elbow			Knee		
	R ²	RMSE (Nm)	NRMSE (%)	R ²	RMSE (Nm)	NRMSE (%)
Individualized Models						
Participant 1	0.91	2.82	7.4	0.93	10.87	6.3
Participant 2	0.86	4.22	10.0	0.92	12.85	6.9
Participant 3	0.96	3.56	6.0	0.94	15.81	7.1
Participant 4	0.93	4.46	6.6	0.94	12.68	6.1
Participant 5	0.87	3.44	9.5	0.93	8.79	6.8
Participant 6	0.94	3.35	5.8	0.92	12.99	7.4
Participant 7	0.93	3.40	6.7	0.96	9.61	4.9
Participant 8	0.91	5.67	7.5	0.90	19.28	10.0

[Updated Text]

Methods

Experimental methods and data processing

To isolate biological torque changes, we first modeled gravitational torque (torque measurements in passive contractions) as a **sinusoidal** function to joint angle and subsequently subtracted this function from torque measurements in all passive and active contractions.

Supplementary Information

Table S1: Estimation accuracies for (MT, Ang)² models in isokinetic dynamometer tests.

Metric	Elbow			Knee		
	R ²	RMSE (Nm)	NRMSE (%)	R ²	RMSE (Nm)	NRMSE (%)
Individualized Models						
Participant 1	0.91	2.83	7.6	0.93	10.95	6.2
Participant 2	0.86	4.22	10.2	0.92	12.76	6.8
Participant 3	0.96	3.56	6.2	0.94	15.77	7.1
Participant 4	0.93	4.47	6.7	0.94	12.62	5.9
Participant 5	0.87	3.45	9.5	0.93	8.78	6.8
Participant 6	0.94	3.39	5.8	0.92	12.97	7.4
Participant 7	0.93	3.40	6.7	0.96	9.57	4.9
Participant 8	0.91	5.67	7.5	0.90	19.23	9.7

- In "Dumbbell curl rigid body model" L is the distance from the elbow to the hand. Since the hand has its own length, this is a generic definition. Authors should better indicate the anatomical reference points used to assess the lengths.

Thank you for your comment. We have revised the writing with clearer anatomical reference points.

[Updated Text]

Methods

Dumbbell curl rigid body model

L is the distance from the cubital fossa (elbow pit) to the center of the third metacarpal bone (center of the hand) at 0° wrist flexion.

- Equation 1 assumes that the centre of gravity lies in the middle of L . Why?

Thank you for your question regarding the rigid body model from the dumbbell curl test. To simplify the mathematical model, we assumed that the forearm and the hand is a rigid rod with uniformly distributed mass. Therefore, its center of gravity lies in the middle of L . We acknowledge that this is a significant simplification and would likely lead to some of the discrepancies in the estimation. We have revised our writing to state this assumption and its potential implications more clearly.

[Updated Text]

Results

Elbow torque estimation during dumbbell curls

Such discrepancy is likely attributed to factors such as muscle co-contraction, which the rigid model cannot capture, as well as the various simplifying assumptions employed in the rigid body model (Methods).

Methods

Dumbbell curl rigid body model

We modeled the forearm and hand as a rigid rod with uniformly distributed mass and the dumbbell as a point mass at the distal end of the rod (Fig. S7).

Reviewer #3

The manuscript “Estimation of Joint Torque in Dynamic Activities Using Wearable A-Mode Ultrasound” reports a dynamic muscle thickness extraction and joint torque estimation method based on wearable A-mode ultrasound. The study proposes a multi-view A-mode ultrasound sensing solution and a hybrid muscle thickness tracking method for robust extraction of muscle thickness in dynamic motions, as well as a second-order polynomial fitting method that integrates joint kinematic information for better joint torque estimation. Exhaustive simulations and experiments were performed to validate the effectiveness of the proposed method, with very promising results achieved. This work will contribute to an in-depth understanding of muscle activity during dynamic and unconstrained movements and holds great promise for biomechanics and rehabilitation robotics research.

Thank you for your accurate and comprehensive overview of our work. We really appreciate your insightful comments and your acknowledgement of the potential impact of our work.

1. The main innovation of this paper is to achieve accurate muscle thickness tracking using wearable A-mode ultrasound through a multi-view sensing strategy and a hybrid muscle thickness extraction algorithm. While the algorithm works well for offline analysis, I am interested in how the algorithm can be used for real-time applications (e.g., real-time estimation of joint torque for robot control). As a lightweight sensing strategy, it will be more interesting to combine A-mode ultrasound with wearable robotics, which requires real-time computational power of the algorithms. Offline analysis is also important for certain biomechanical studies, however, B-mode ultrasound can also work well in this case.

We thank the reviewer for their interest in the real-time aspect of our algorithm. We completely agree with the reviewer that wearable robotics is a very exciting field and could be a great application for our work if it could be implemented in real-time. Although the MBTA and torque estimation are currently conducted in post-processing, we believe in their potential for real-time implementation. In response to your comment, we have updated our writing to highlight the potential of our work for real-time applications.

[Updated Text]

Discussion

Third, while the MBTA demonstrates potential for real-time implementation (Methods), current torque estimation is conducted in post-processing. Future efforts could focus on developing a centralized system to simultaneously log ultrasound and IMU data and perform the torque estimation algorithm in real-time.

Methods

Muscle boundary tracking algorithm (MBTA)

Although the MBTA is currently implemented in post-processing, its required computational steps are straightforward. The algorithm operates with an average runtime of less than 0.3 milliseconds per frame on a standard CPU (Intel Core i7-10750H @ 2.60 GHz), showcasing its potential for real-time use. To transition to real-time implementation, further developments could involve pre-defining the region of interest based on body measures, such as weight and height, replacing low-pass filters with the simpler moving average filters, and implementing the algorithm on a field programmable gate array (FPGA). Notably, the 2D box filter in the brightness-based method and the smoothing filters in

the final fusion step may introduce time delays in tracking. Therefore, it is important to optimize filter parameters, like filter size and cutoff frequency, for specific applications to minimize these delays.

2. Regarding the muscle boundary tracking algorithm, the authors state that the sensor with the strongest echoes is used when measuring muscle thickness. This sensor selection method helps to select the sensor with the least rotation to the target muscle. However, during the muscle contraction, the deformation of the muscle changes the relative angle between the muscle and the sensor, which means that the optimal sensing angle is constantly changing during muscle contraction. How did the authors solve this problem? For offline analysis, some filtering methods can help smooth out the results, even if there is some jitter. However, this can cause problems for online applications. In terms of the simulation study in Fig. S3, what is the amplitude of the sine wave?

We thank the reviewer for their carefully inspection and pointing out the potential confusion in our writing. The reviewer is completely right that the angle between the target muscle boundary and the transducer constantly changes during active contraction. We addressed this issue with the MBTA by fusing outputs from two different tracking methods to provide robust muscle thickness tracking during motion. Our original writing might have been unclear in delivering this point. We have revised our writing to make it clearer.

Yes, filtering techniques, like low-pass filters, can effectively remove the motion-induced jitters in thickness tracking. Although our current implementation of the MBTA is in post-processing, as mentioned in our response to your previous comment, we believe it has the potential to be implemented in real-time. And the moving average filter can be a real-time alternative to low-pass filters, but with a small time delay.

The sine wave in Fig. S3 has amplitudes ranging from 1 to 20 to simulate the 95% signal drop. We have revised Fig. S3 by adding the proper y-axis label.

[Old Text]

Results

Muscle thickness tracking during motion

To enable joint torque estimation, we developed a method to robustly measure muscle thickness using A-mode ultrasound. We designed a transducer mount capable of holding four SETs at different angles (Fig. 1a), which can be worn on the limb above the target muscle (Fig. 1b, Fig. 1c). Additionally, we developed a custom muscle boundary tracking algorithm (MBTA) that identifies the transducer with the strongest echoes and measures changes in muscle boundary depth and muscle thickness (Fig. 1d). We designed the transducer mount and the MBTA to account for the dependence of ultrasound echo intensity on the angle of incidence. Specifically, the ultrasound signal strength is maximized when the transducer faces parallel to the boundary of interest, but quickly degrades with misalignments (Fig. S2). In practice, the angle between the skin and the muscle boundary differs among participants and changes during motion. To mitigate the effect of such angle variation, we implemented the sensor redundancy to accommodate the different muscle shapes among participants and designed the MBTA to withstand motion-induced variations in echo intensity (Fig. S1).

[Updated Text]

Results

Muscle thickness tracking during motion

To enable joint torque estimation, we designed a transducer mount capable of holding four SETs at different angles (Fig. 1a), which can be worn on the limb above the target muscle (Fig. 1b, Fig. 1c). This sensor redundancy was implemented to account for the difference in muscle shape among participants. Specifically, ultrasound echo intensity is highly dependent on the angle of incidence, with the intensity maximized when the ultrasound beam hits the target boundary at a 90° angle (Fig. S2). With multiple transducers at a range of angles, we increased the likelihood of normal incidence while collecting from muscles with different sizes and shapes.

To measure muscle thickness, we developed a custom muscle boundary tracking algorithm (MBTA) that identifies the transducer with the strongest echoes and uses its signal to measure muscle boundary depth (Fig. 1d). The MBTA can continuously track thickness changes in both superficial and deep muscles during contractions (Fig. S1). In practice, muscle deformation constantly alters the angle between its boundary and the transducer, especially during dynamic motion. To address the resultant variations in ultrasound echo intensity, the MBTA fuses results from two different tracking methods to provide robust estimation of muscle thickness (Supplementary Text). To test the MBTA's tracking performance and robustness, we...

3. For the comparison of A-mode and B-mode ultrasound for muscle thickness tracking in Fig. S4, the processing of B-mode ultrasound is still in an A-mode way. The authors selected some columns from the B-mode ultrasound image to simulate the use of A-mode ultrasound sensing. Since it is comparative analysis here, why not extract the muscle thickness from the B-mode to provide a better ground truth?

Thank you for your thorough review of our experiment. We agree with the reviewer that it would be preferable to use an image-based technique for processing the B-mode data than to simulate A-mode data from it. In response, we have reprocessed the B-mode data by finding the 2D muscle boundary within each B-mode ultrasound image and measuring the thickness over time at the location corresponding to the SET placement. As shown below, this updated processing technique did not drastically change the comparison results.

[Old Text]

Results

Muscle thickness tracking during motion

Compared to the muscle thickness measured with B-mode, A-mode ultrasound achieved an RMSE of 0.46 mm and a NRMSE of 1.6%, ...

Supplementary Information

Comparing muscle thickness tracking using A-mode and B-mode ultrasound

In post-processing, we extracted a vertical line from each B-mode image, specifically from a column near the SET location (Fig. S4c). We then concatenated these lines to form a signal resembling A-mode ultrasound data and applied the MBTA to determine the ground truth muscle boundary depth (Fig. S4d).

[Updated Text]

Results

Muscle thickness tracking during motion

Compared to the muscle thickness measured with B-mode, A-mode ultrasound achieved an RMSE of 0.48 mm and a NRMSE of 1.7%, ...

Supplementary Information

Comparing muscle thickness tracking using A-mode and B-mode ultrasound

In B-mode processing, we first cropped ultrasound images to isolate the target muscle boundaries. We then found the 2D muscle boundary from each B-mode image by identifying a line spanning across the full width of the image and produces the highest mean pixel intensity (Fig. S4c). Lastly, we extracted a single depth value from the identified line at the location corresponding to SET placement, recorded it over time, and used the resulting time-series data as the ground truth muscle thickness (Fig. S4d).

[Updated Figure]

Fig. S4: Comparison of A-mode and B-mode ultrasound for muscle thickness tracking. (A) Illustration of the experimental setup. (B) Photograph of a 3D printed case that can mount up to eight A-mode SETs and a B-mode linear array transducer (LAT). (C) Identification of 2D muscle boundary (green dashed line) using the B-mode image at each time frame. (D) B-mode measured time-series muscle thickness plots obtained by extracting depth values from the identified muscle boundaries near the SET placement (white vertical dashed line in C). (E) A-mode ultrasound data with MBTA tracing (green dashed line) overlaid. (F) Time-series error plots comparing A-mode-based with B-mode-based muscle thickness measurements. Red lines represent the RMSE of 0.48 mm.

4. There are quite a few single case experiments in the paper, such as dumbbell curls and treadmill/outdoor locomotion tests. These experiments make more sense to the reader, and it would be nice to extend them a bit. In particular, for the dumbbell curl experiment (Fig. 3), I am supervised that the elbow torque estimated by the A-mode ultrasound matched the simple simulation model quite well.

We thank the reviewer for their comment to extend the participant pool for the dumbbell curl and the locomotion tests. We acknowledge the lack of participants for making stronger claims regarding the method's effectiveness in unconstrained dynamic activities. In response, we have increased the number of participants in the dumbbell curl and treadmill locomotion tests to 5.

We have maintained N=1 for the outdoor locomotion test. We justify this choice based on the absence of ground truth in outdoor environments and the primary objective of this test, which is to demonstrate applicability in real-world settings rather than quantitative validation of the proposed method (which is the goal for the treadmill part with N=5).

We refer the reviewer to the revised manuscript with highlighted changes for a comprehensive overview of the changes but have included a few call outs below. Also, we were surprised by the match in the dumbbell curl experiment as well and now show similar trends in the other 4 new participants in the newly added Fig. S8.

[Updated Text]

Results

Elbow torque estimation during dumbbell curls

To investigate functional applications of elbow torque estimation, we performed a study on five participants during dumbbell curls.

...

For all participants, there was close agreement between the estimated and calculated torque (Fig. S8), with the average elbow torque increasing proportionally to the weight of the dumbbell (Fig. 3e, Fig. 3f). Quantitatively, absolute errors in average torque measurements were 1.25 ± 0.71 Nm for no weight, 1.18 ± 0.57 Nm for medium weights, and 2.41 ± 0.94 Nm for heavy weights across all participants. Qualitatively, ...

Knee torque estimation during treadmill and outdoor locomotion

To evaluate knee torque estimation during unconstrained dynamic activities, we performed a treadmill study on five participants and a single-participant outdoor locomotion demonstration.

...

By fitting quadratic fits on data from each participant across all treadmill conditions, we obtained individualized models with RMSEs of 17.43 ± 5.09 Nm, NRMSEs of $6.0 \pm 1.1\%$, and R^2 of 0.92 ± 0.03 (table S2).

[Updated Figures]

Results

Elbow torque estimation during dumbbell curls

(We included the following subfigures in Fig. 3 to show the results from all 5 participants.)

Fig. 3. Elbow torque estimation during dumbbell curls. (E) Average estimated torque for all participants ($n = 5$) at different weight conditions. Each color represents data for one participant. Each dot is obtained by taking the average of all contractions ($n = 6$) within the respective condition. (F) Average calculated torque for all participants ($n = 5$) at different weight conditions.

Knee torque estimation during treadmill and outdoor locomotion

(We updated Fig. 5c and Fig. 5d to show the effects of locomotion type and slope level on RMSEs for all 5 participants. Also, we changed the plots from bar plots to box plots to better show the underlying data distribution.)

Fig. 5. Knee torque estimation during treadmill and outdoor locomotion. (C) Distributions of RMSEs for different locomotion types across all participants (n = 5). Colored dots represent the RMSE for each participant. (D) Distributions of RMSEs for different slope levels across all participants (n = 10).

Supplementary Information

(We added the following supplementary figure to show data for all participants during the dumbbell curl test.)

Fig. S8: Data from different participants during the dumbbell curl study. Each row represents BB thickness (left column), elbow angle (center left column), estimated elbow torque (center right

column), and calculated elbow torque from a rigid body model (right column) for one participant during curls with different dumbbell weights. Data from Participant 1 is shown in Fig. 3. Lines and shaded regions represent mean \pm SD (n = 6 repetitions).

5. In terms of second-order polynomial fitting, it is basically similar to the linear regression applying a second-order polynomial basis function. Incorporating joint kinematics into the model also makes sense because both muscle activation (thickness in this case) and muscle length (joint kinematics) affect the joint torque. When building the polynomial fitting model, did the authors use the data covering all the different contraction tasks? If so, the model might be overfitted because all the data came from repeatable movements. It is better to test the generalisability of the model by training and testing it on different contraction tasks.

Thank you for your insightful comment. Indeed, in the dynamometer tests (Fig. 2, Fig. 4), we generated quadratic models using data from all contraction tasks. We used all available data in these two experiments because our goal was to explore whether simple and descriptive relationships could be established from muscle deformation to joint torque, rather than to propose a new ML-based estimation algorithm. Notably, this technique of assessing relationships using all available data is commonly employed in similar biomechanical studies, such as:

- Martin, Jack A., et al. "Gauging force by tapping tendons." *Nature communications* 9.1 (2018): 1592.
- Alvarez, Jonathan T., et al. "Towards soft wearable strain sensors for muscle activity monitoring." *IEEE Transactions on Neural Systems and Rehabilitation Engineering* 30 (2022): 2198-2206.
- Hodges, P. W., Pengel, L. H. M., Herbert, R. D. & Gandevia, S. C. Measurement of muscle contraction with ultrasound imaging. *Muscle Nerve* 27, 682–692 (2003).

Yet, in the dumbbell curl and outdoor locomotion tests, we trained and evaluated the model using data from different contraction tasks (Fig.3, Fig. 5f, Fig. 5g). Specifically, we trained quadratic fits with data collected on the dynamometer and treadmill and subsequently evaluated them during unconstrained dumbbell curls and outdoor locomotion activities.

Moreover, in our experiments, the number of data points significantly exceeded the tunable parameters in quadratic models. For example, for one participant from the dynamometer test, the (MT, Ang)² model used 10858 data points to determine 6 tunable coefficients in the model. Given this substantial data-to-parameter ratio, we were less concerned about the risk of model overfitting. Additionally, we have incorporated two new references to support our claim that simpler models enhance generalizability and transferability.

However, we recognize that our original writing might not have clearly conveyed these points. We have revised the relevant section in Discussion to articulate these arguments more clearly and to suggest that incorporating more advanced machine learning algorithms could be a promising direction for future research. Once again, we thank the reviewer for helping us to strengthen the paper.

[Old Text]

Discussion

To describe the relationship between input measures (muscle thickness, joint angle) and joint torque, we used a simple 2nd order polynomial fit, which offers advantages in preventing overfitting and

improving model generalizability. A strength of our estimation was the robustness across different contraction speeds (Fig. 2d, Fig. 4d) and types (concentric vs eccentric) (Fig. 2e, Fig. 4e). Our estimation also showed generalizability across participants, as our generalized models produced less than 4% additional NRMSEs compared to individualized models for both elbow and knee torque estimation (Table S1). In contrast, prior A-mode ultrasound studies have primarily used more advanced machine learning algorithms, such as deep neural networks, for various classification applications^{38,40–42} and continuous joint kinematics⁶¹ or static grip force^{43,44} estimation. These algorithms automatically learn salient features from A-mode ultrasound data and map them to target variables. However, their training processes often require substantial amounts of data due to the large number of tunable parameters within the model. In comparison, we alleviated the data requirement by introducing the MBTA and applying a relatively simple quadratic fit on the extracted muscle thickness, rather than on raw A-mode ultrasound data. As an example, in the dumbbell curl demonstration, we constructed the elbow torque estimator using only 15 isokinetic contractions (5 passive, 5 concentric, 5 eccentric). We expect that combining a feature extraction method, like the MBTA, with advanced machine learning algorithms could further improve estimation accuracy in the future, potentially by fusing multiple measurements from the same muscle or leveraging measurements from different muscles.

[Updated Text]

Discussion

To describe the relationship of muscle thickness and joint kinematics to joint torque, we used a simple 2nd order polynomial fit, which offers advantages in model generalizability and transferability due to its simple architecture^{66,67}. Specifically, we showed in the dynamometer tests that applying the quadratic fit to all contractions yielded comparable results across different contraction speeds (Fig. 2d, Fig. 4d) and types (concentric vs eccentric) (Fig. 2e, Fig. 4e). We further showed in the dumbbell curl and the outdoor locomotion tests that quadratic models trained using controlled contractions can generate promising results in unconstrained dynamic tasks (Fig. 3c, Fig. S8, Fig. 5f, Fig. 5g). In contrast, prior A-mode ultrasound studies have primarily used more advanced machine learning algorithms, such as deep neural networks, by training them directly on the raw ultrasound data. Despite enabling various classification applications^{40,42–44} and continuous joint kinematics⁶⁸ or static grip force^{45,46} estimation, these algorithms often require substantial amounts of training data due to the large number of tunable parameters within the model. In comparison, we alleviated the data requirement by training the relatively simple quadratic fit on the extracted muscle thickness data, rather than raw A-mode signals. As an example, we trained the elbow torque estimator for dumbbell curls using only 15 isokinetic contractions (5 passive, 5 concentric, 5 eccentric). In the future, we expect that combining a feature extraction method, like the MBTA, with advanced machine learning algorithms could enable more accurate torque estimation, potentially by fusing multiple measurements from the same muscle or leveraging measurements from different muscles.

References

66. Forster, M. R. Key Concepts in Model Selection: Performance and Generalizability. *J. Math. Psychol.* 44, 205–231 (2000).
67. Lute, A. C. & Luce, C. H. Are Model Transferability And Complexity Antithetical? Insights From Validation of a Variable-Complexity Empirical Snow Model in Space and Time. *Water Resour. Res.* 53, 8825–8850 (2017).

6. In the locomotion test, the authors took the hip-knee angle difference and the thigh angle input

kinematic variables. What is the difference between thigh angle and hip angle? I recommend inputting hip and knee angles into the model because the model will handle the difference if necessary.

We thank the reviewer for pointing out this potential confusion. Hip angles can be calculated by taking the difference between pelvis and thigh angles, which are the segment angles measured by the pelvis and thigh IMUs respectively. We acknowledge that these terms may not have been clearly defined in the original submission. Therefore, we have revised the Methods section to provide clearer definitions.

We also agree with the reviewer that the quadratic fit should be capable of conducting subtractions among the different angles if necessary. To have more intuitive input variables, we have updated the kinematic inputs to the model from the original “hip-knee angle difference and thigh angle” to “pelvis, thigh, and shank angles” instead. We have also updated all the results in reflection of this change.

[Old Text]

Results

Proof of concept: Knee torque estimation during treadmill and outdoor locomotion

The fitted quadratic model used three inputs: RF thickness, hip-knee angle difference, and thigh angle.

Methods

IMU-based kinematics calculation and gait segmentation

At each time frame, we measured body segment angles (angles of the forearm, upper arm, torso, thigh, shank, and foot relative to gravity) by redefining the IMU global frame (with one axis aligned with the joint axis and another pointed against gravity) and finding the pitch angle between the local and the updated global frames around the joint axis. We then calculated joint angles (elbow, knee, and hip angles) by subtracting the angles of two adjacent segments (e.g., knee angle = thigh angle – shank angle).

[Updated Text]

Results

Knee torque estimation during treadmill and outdoor locomotion

These results were generated using quadratic models with RF thickness, along with pelvis, thigh, and shank angles (directly measured by IMUs) as input variables. These input variables were used to capture the combined effect of muscle deformation and joint kinematics on knee torque estimation.

Methods

IMU-based kinematics calculation and gait segmentation

With these assumptions, each IMU can measure the angle of respective body segment (forearm, upper arm, pelvis, thigh, shank, and foot) relative to gravity. Specifically, we measured these segment angles by redefining the IMU global frame (with one axis aligned with the joint axis and another pointed against gravity) and finding the angle between the local and the updated global frames around the joint axis. Joint angles can then be calculated by subtracting the angles of two adjacent segments (e.g., hip angle = pelvis angle - thigh angle, knee angle = thigh angle - shank angle).

7. In Fig.1 D, the red line doesn't appear to match the strongest echo in the signal.

Thank you for your detailed observation. The MBTA tracks the depth of muscle boundaries by using a short window of data around each time frame, rather than solely relying on the data from that specific time frame (Methods). As a result, the MBTA measurement at any given time frame may not necessarily align to the strongest echo in the ultrasound signal. We acknowledge that our previous Fig. 1d may have been confusing. In response, we have updated the figure by removing the red lines from the insets and now use these insets solely for visualizing the raw A-mode waveforms.

REVIEWERS' COMMENTS

Reviewer #1 (Remarks to the Author):

The authors addressed my concerns, they even increased the sample size, so I am satisfied about the work done in the revised version of the manuscript

Reviewer #2 (Remarks to the Author):

Ergonomics

Enthesopathy is a major occupational disease. Authors decided to focus on epicondylitis. It would be useful to mention the prevalence of this condition, so to better assess the potential impact of the technology in ergonomics.

A simple mention in the discussion is quite lateral. Please mention the potential application in ergonomics since the abstract/introduction.

MMG

Quite dated references are provided (2012, 2014) to support the conclusion that "MMG is highly susceptible to motion artifact from limb movement, especially during dynamic activities". Compensation strategies, involving the multimodal use of additional sensors (e.g. accelerometers) is not mentioned here. The advantage of the proposed sensory system is not thoroughly explained.

Resistive elastic bands

Authors state: “We believe that ultrasound holds a unique advantage of providing muscle specific measurements”. The answer would be more convincing if the proposed technology were compared with alternative, multimodal sensing technologies.

Real-time use

OK

Use of different ultrasonic elements

OK

Number of participants to validate the algorithm

OK

How the algorithm works in the case of deeper muscles or in the case of ball-socket joints

OK

Empirical approach

Replacing $(MT+Ang)^2$ with $(MT, Ang)^2$ is but a cosmetic change. Authors should list possible biomechanically grounded models, compare them, and show how/why the model they propose is to be preferred.

Reviewer #3 (Remarks to the Author):

The authors have addressed my questions carefully, and I have no more questions now.